# Dynamic Compression Flows for Neuroscience Data

Ganchao Wei [* 1 2]   Daniela de Albuquerque [* 3 4]   Miles Martinez [3]   Shiyang Pan [3]   John Pearson [1 3]

## Abstract

While neuroscience experiments have repeatedly demonstrated the involvement of large populations of neurons in even simple behaviors, these studies have just as often reported that the collective dynamics of neural activity are approximately low-dimensional. As a result, methods for identifying low-dimensional latent representations of time series data have become increasingly prominent in neuroscience. However, most existing methods either ignore temporal structure or model time evolution using latent dynamical systems approaches. In the first case, dynamics may be distorted or even scrambled in the latent space, while in the second, many possible latent dynamics may give rise to the same data. Here, we address these challenges using a novel flow-matching approach in which data are generated by a pair of flow fields, one governing time evolution, the other a mapping between data and a low-dimensional latent space. Importantly, the dimension-reducing flow is trained to minimize distortions of the temporal dynamics, learning an identifiable low-dimensional representation that preserves temporal relations in the original data. Additionally, we constrain our latent spaces to have low-dimensional support in a soft, parameterized manner, taking inspiration from ideas on nested dropout. Across both neural and behavioral data, we show that this dual flow approach produces both more interpretable dynamics and higher-quality reconstructions than competing models, including in noise-dominated data sets where conventional approaches fail.

## 1. Introduction

The emergence of large-scale neural recording technologies has drastically changed our understanding of neural function, shifting systems neuroscience from a single unit perspective to a focus on neural populations and their collective dynamics. Fortunately, several lines of empirical evidence have shown that such seemingly complex and high-dimensional data can actually be described in terms of a much smaller number of "latent" variables (Gao et al., 2017; Trautmann et al., 2019; Vyas et al., 2020; Ebitz & Hayden, 2021). Not surprisingly, this observation has led to a proliferation of dimensionality reduction algorithms for neuroscience data.

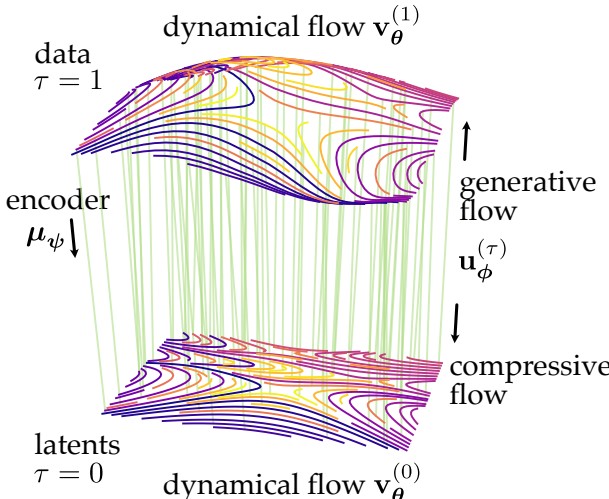

*Figure 1.* **Dynamic Compression Flows for dimension reduction**. Dynamical data $\mathbf{x}_t$ at $\tau = 1$ with dynamics defined by $\mathbf{v}_\theta$ are mapped to a lower-dimensional compressed representation ($\tau = 0$) via a compressive/generative flow $\mathbf{u}_\phi$. Both $\mathbf{u}_\phi$ and $\mathbf{v}_\theta$ are trained via flow matching defined by an encoder/coupling $\boldsymbol{\mu}_\psi$.

These algorithms broadly fall into two major types: In the first, the primary goal is to construct a lower-dimensional representation of the original high-dimensional data while maximizing some measure of information retained. This "mapping" from data to latent space can be accomplished using classic linear (e.g., PCA, ICA, NMF, Mairal et al., 2009; Charles et al., 2011b; Draelos et al., 2021), as well

---

[*]Equal contribution  [1]Department of Neurobiology, Duke University, Durham, NC, USA [2]Department of Statistical Science, Duke University, Durham, NC, USA [3]Department of Electrical and Computer Engineering, Duke University, Durham, NC, USA [4]School of Medicine, Duke University, Durham, NC, USA. Correspondence to: Ganchao Wei <ganchao.wei@duke.edu>.

*Proceedings of the 43rd International Conference on Machine Learning*, Seoul, South Korea. PMLR 306, 2026. Copyright 2026 by the author(s).

as non-linear approaches (e.g., Isomap, LLE, Tenenbaum et al., 2000; Roweis & Saul, 2000), visualization-based approaches (e.g., t-SNE, UMAP, PHATE, Van der Maaten & Hinton, 2008; McInnes et al., 2018; Moon et al., 2019), and even more modern approaches such as VAEs (Kingma & Welling, 2014; Rezende et al., 2014; Goffinet et al., 2021; Martinez & Williams, 2026), self-supervised learning (Azabou et al., 2024; Schneider et al., 2023), and the information bottleneck (Tishby et al., 2000). Regardless of the specific method, such models often ignore temporal structure, treating data as independent. Consequently, learned representations will often severely distort and scramble the original dynamics.

In the second class of models, nonlinear dynamical systems approaches attempt to capture temporal evolution directly, fitting flow fields to data (Charles et al., 2011a; Kutz et al., 2016; Gao et al., 2016; Rajan et al., 2016; Linderman et al., 2017; Pandarinath et al., 2018; Kerg et al., 2019; Zhao & Park, 2020; Wiltschko et al., 2020; Nair et al., 2023; Busch et al., 2023; Driscoll et al., 2024; Weinreb et al., 2024). Despite their considerable success in identifying repeating dynamical motifs in data (Mante et al., 2013; Markowitz et al., 2023; Nair et al., 2023; Liu et al., 2024; Vinograd et al., 2024), such models still struggle to identify useful structure under (1) non-repeatability and (2) noise-dominant regimes, which are ubiquitous in neuroscience data. That is, when modeling data without the benefit of smoothing and trial-averaging, these methods can struggle to identify structure, since they are tailored to identify repeating dynamical motifs (Williams & Linderman, 2021). Likewise, most models assume that noise is small and/or follows simple forms, so that system evolution is governed by a well-defined velocity field. However, these assumptions can fail catastrophically when variance in the data is mostly due to unmeasured variables (Musall et al., 2019) or has heavy-tailed structure, resulting in latent dynamics that appear random rather than lawful (Draelos et al., 2021).

Recent breakthroughs in "simulation-free" flow training and the development of flow matching (Lipman et al., 2024; 2023; Albergo & Vanden-Eijnden, 2023; Albergo et al., 2025; Pooladian et al., 2023; Tong et al., 2024a) from earlier diffusion-based models (DBMs, Song et al., 2021; Karras et al., 2022) have led to an explosion of work using latent flows (Polyak et al., 2024; Dao et al., 2023; Hu et al., 2024; Schusterbauer et al., 2024) and diffusions (Vahdat et al., 2021; Blattmann et al., 2023; Preechakul et al., 2022; Hudson et al., 2024). Yet despite their huge success as generative models, most of these approaches do not directly allow for dimensionality reduction, relying instead on a front-end encoder network to infer latent representations from data. Unfortunately, this approach can re-introduce the same identifiability issue mentioned above. Even though recent diffusion-based models for neural latent dynamics

and spiking data do address low-dimensional latent structure (Wang et al., 2023; Kapoor et al., 2024), our focus here is instead on using flow matching to construct dynamics-preserving representations of data while also directly learning the velocity fields. Additionally, in standard diffusion formulations, the source distribution is typically fixed to be Gaussian, while flow matching can define transport between arbitrary source and target distributions, which may be helpful in understanding non-Gaussian neural activity distributions across time.

Here, we propose *Dynamic Compression Flows* (DCFs) as a means of inferring low-dimensional latent structure in a way that respects temporal dynamics in the data (**Figure 1**). **Our contributions are as follows:**

1. We develop a *dual* flow-matching approach, learning one generative/compressive flow field that maps the data to a low-dimensional latent space and another that captures temporal dynamics at each level of compression. Critically, the latent representations inferred by our model remain *identifiable* (up to a sign), making latent spaces reproducible across runs and thereby addressing a major limitation of previous approaches.

2. We achieve low-dimensional support for our latent distribution within the embedding space by training using nested dropout (Rippel et al., 2014), which ensures that our latent dimensions are ordered by construction while also allowing for *controllable* and *soft* dimensionality reduction.

3. We apply our proposed model extensively to both synthetic and benchmark neural and behavioral data and compare it against a variety of competing approaches, demonstrating both its effectiveness and superior performance in challenging, noise-dominated regimes.

**Conflict of Interest Disclosure: The authors have no conflict of interest to disclose.**

**Related work: Flow-Based Approaches in Latent Space.** Historically, *Injective Flows* (Brehmer & Cranmer, 2020; Cornish et al., 2020; Flouris & Konukoglu, 2023; Caterini et al., 2021) were among the first flow-based approaches aimed at learning one-to-one mappings (i.e., injections) between points in data space and points in a lower-dimensional latent space. However, early formulations of these models suffered from low expressivity due to the need for efficient Jacobian determinant calculations. More recently, as reviewed above, "simulation-free" training approaches have led to an explosion of latent flow-based models, but most such approaches still rely on pre-trained encoders for initial compression and are focused on generative performance as opposed to inference. Parallel lines of work have also sought

to construct flow-based models that respect data manifold geometry (Mathieu & Nickel, 2020; Lou et al., 2020; Falorsi, 2021; Chen & Lipman, 2024; Atanackovic et al., 2025; Kapusniak et al., 2024; Kruiff et al., 2024; Diepeveen et al., 2025). However, this work has focused on generation on manifolds *per se*, not on dimension-reduced representations or latent dynamics.

Perhaps the closest approach to ours is de Albuquerque & Pearson (2024), who used probability flow ODEs (pfODEs) to directly map high-dimensional data into an effectively lower-dimensional latent space, while preserving identifiability. Importantly, that work *did not* model temporal dynamics. Additionally, that work proposed a "PR-reducing" noise schedule to map data into a low-rank source Gaussian. Here, we opt instead for additional flexibility, and use a *jointly trained* encoder network to learn our couplings, effectively allowing for more flexible source distributions. Another closely related work (Cai et al., 2025) proposed constructing a dual conditional flow-matching formulation to map data into a *learned lower dimensional* latent space and back. That work differs from this one in two important ways: First, it does not seek to model latent dynamics. Second, it uses a particular form of prior-informed coupling rather than learning the coupling from data, as we do.

**Related work: Dynamical Modeling.** Neural ODEs and SDEs have been extensively used to model temporal dynamics and irregularly sampled time series data (Chen et al., 2018; Rubanova et al., 2019; Brouwer et al., 2019; Dupont et al., 2019; Hasan et al., 2022; Cheng et al., 2025). Historically, training such models required expensive backpropagation through the SDE/ODE solver, greatly restricting the scalability of such methods. However, recent breakthroughs in score and flow matching have allowed for efficient *simulation-free* training of such models (Zhang et al., 2024; Bartosh et al., 2025; Tong et al., 2024b). Additionally, several lines of work have extended existing score and flow matching approaches to multi-marginal settings (Albergo et al., 2024; Lee et al., 2025; Rohbeck et al., 2025) with the goal of modeling complex dynamics from limited "snapshot" observations sampled at irregular time points. In parallel, Wei & Ma (2025) proposed constructing conditional probability paths along instances of latent stochastic paths (i.e., "streams") and modeling such streams using Gaussian Processes (GPs, Rasmussen & Nickisch (2010)), effectively creating a new GP-based flow-matching model capable of handling multiple correlated training points (i.e., a time series). Other lines of work have also extended flow-matching to model time point processes (TPPs, Shou, 2025; Kerrigan et al., 2024; Mukherjee et al., 2025), however the focus there is instead on event timing and occurrence (i.e., forecasting), as opposed to state evolution throughout time. Of note, none of these approaches seek to directly infer a latent

representation of the data that preserves intrinsic dynamics, a key goal in neuroscience.

## 2. Model

### 2.1. Notation

Let $\mathbf{x}_t \in \mathbb{R}^D$ denote the observation at time $t$. We use a superscript $(\tau)$ for the *compression coordinate*, where $\tau = 1$ is data space and $\tau = 0$ is the compressed space. Therefore, $\mathbf{x}_t^{(\tau)}$ denotes the state at compression level $\tau \in [0, 1]$ and time $t$, so that $\mathbf{x}_t^{(1)} = \mathbf{x}_t$. In the following, we assume that data are sampled at times $t = k\Delta t$, $k \in \mathbb{Z}$, though our approach can accommodate non-uniform spacing. Our goal is to learn a pair of flow fields: a compressive flow $\mathbf{u}_{\phi}$ that transports states along $\tau$ from a low-dimensional generative subspace to high-dimensional data space, and a dynamical flow $\mathbf{v}_{\theta}$ that transports states through time within each $\tau$.

### 2.2. Encoder

For training a flow-matching model, we require a mapping, called the *coupling*, for pairing points in the target (data) distribution $p_{\tau=1}(\mathbf{x})$ with those in the source (latent) distribution $p_{\tau=0}(\mathbf{x})$ (Lipman et al., 2024). Here, rather than fixing the form of the source distribution, we instead choose a deterministic coupling that enforces dimension reduction:

$$\mathbf{x}_t^{(0)} = \mathbf{b} + \mathbf{L}\mathbf{D}^{1/2} \cdot \boldsymbol{\mu}_{\boldsymbol{\psi}}\left(\mathbf{x}_t^{(1)}\right), \qquad (1)$$

where $\mathbf{b} \in \mathbb{R}^D$ is a bias term, $\mathbf{L} \in \mathbb{R}^{D \times D}$ has orthonormal columns (e.g., parameterized via non-pivoted QR), $\mathbf{D} = \mathrm{diag}(d_1, \ldots, d_D)$ with $d_i > 0$, and $\boldsymbol{\mu}_{\boldsymbol{\psi}}(\mathbf{x}) \in \mathbb{R}^D$ is a parameterized nonlinear mapping.

This parameterization separates source distribution orientation and spread: $\mathbf{L}$ defines the axes of the linear subspace in ambient space, and $\mathbf{D}$ controls per-coordinate scale, since sample energy $\|\mathbf{x}_t^{(0)} - \mathbf{b}\|_2^2 = \boldsymbol{\mu}_{\boldsymbol{\psi}}(\mathbf{x}_t^{(1)})^{\top} \mathbf{D} \, \boldsymbol{\mu}_{\boldsymbol{\psi}}(\mathbf{x}_t^{(1)})$ when $\mathbf{L}$ has orthonormal columns. Moreover, when $\mathbf{D}$ is low-rank, the source points $\mathbf{x}_t^{(0)}$ lie in a low-dimensional subspace. Our goal is to learn an encoding map that effectively minimizes the rank of $\mathbf{D}$ while maintaining the predictive accuracy of both compressive and dynamical flows.

In practice, to stabilize training and remove the scale ambiguity between $\mathbf{D}$ and $\boldsymbol{\mu}_{\boldsymbol{\psi}}(\mathbf{x})$, we normalize the encoder features dimension-wise (Ioffe & Szegedy, 2015; Ba et al., 2016). That is, if $\boldsymbol{\mu}_{\boldsymbol{\psi}}^{\mathrm{raw}}(\mathbf{x}) \in \mathbb{R}^D$ denotes the raw feature output, we set

$$\left[\boldsymbol{\mu}_{\boldsymbol{\psi}}(\mathbf{x})\right]_i = \frac{\left[\boldsymbol{\mu}_{\boldsymbol{\psi}}^{\mathrm{raw}}(\mathbf{x})\right]_i - m_i}{\sigma_i}, \qquad i = 1, \ldots, D, \quad (2)$$

where $\mathbf{m} \in \mathbb{R}^D$ and $\boldsymbol{\sigma} \in \mathbb{R}^D$ are the running (EMA) estimates of the coordinate-wise mean and standard deviation of

the $\boldsymbol{\mu}_{\psi}^{\text{raw}}(\mathbf{x})$, respectively. Thus each coordinate of $\boldsymbol{\mu}_{\psi}(\mathbf{x})$ has approximately zero mean and unit variance over the data distribution. We additionally cap the per-sample $\ell_2$ norm of the raw features to prevent rare large activations from destabilizing the running statistics and downstream flow fitting (Pascanu et al., 2013).

Lastly, and critically for latent space reproducibility, the combination of our parameterization (1), the deterministic non-pivoted QR parameterization for $\mathbf{L}$, the normalization (2), and nested dropout (Section 3.1), which enforces an ordered, prefix-based notion of coordinate importance, renders the encoder identifiable up to a sign for each component.

## Interpolation Scheme

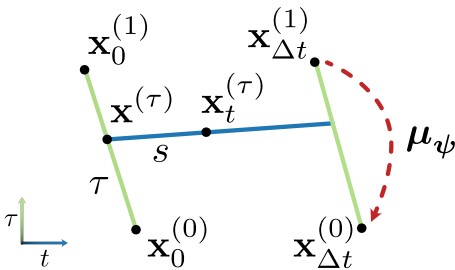

*Figure 2.* **Interpolation scheme for the conditional dynamic flow** (4) **with** $k = 0$. For $\tau, t \in [0, 1]$, intermediate points are first interpolated in the compressive ($\tau$) dimension for each data sample using (3) (green lines), then along the dynamical ($t$) dimension (blue line). The encoder $\boldsymbol{\mu}_{\psi}$ (red line) maps points in the data space ($\tau = 1$) to points in source space ($\tau = 0$).

### 2.3. Compressive Flow

At any fixed time $t$, the compression/generation axis connects a compressed representation to a point in the data. Flow matching learns a marginal probability path by integrating over conditional probability paths (bridges) that connect endpoints drawn from the coupling. Given a data point $\mathbf{x}_t^{(1)}$ and its image under the encoder (1), $\mathbf{x}_t^{(0)}$, we define the linear $\tau$-bridge as

$$\mathbf{x}_t^{(\tau)} = (1 - \tau)\,\mathbf{x}_t^{(0)} + \tau\,\mathbf{x}_t^{(1)}, \tag{3}$$

and learn a compression velocity field

$$\mathbf{u}_{\phi} : \ (\mathbf{x}_t^{(\tau)}, \tau) \ \mapsto \ \partial_{\tau}\mathbf{x} \in \mathbb{R}^D.$$

Note that while the bridge (3) is constructed using the encoder endpoint $\mathbf{x}_t^{(0)}$, the learned flow $\mathbf{u}_{\phi}$ defines a *distinct* projection to $\tau = 0$ via integration from $\mathbf{x}_t^{(1)}$. We denote the data at compressive level $\tau$, obtaining via compressive flow integration, by $\tilde{\mathbf{x}}_t^{(\tau)}$. The flow-based endpoint ($\tilde{\mathbf{x}}_t^{(0)}$) can differ from the encoder endpoint ($\mathbf{x}_t^{(0)}$), effectively rearranging

the encoder geometry at $\tau = 0$. In practice, flow matching tends to produce near-straight $\tau$ paths and local neighborhoods in data space are transported smoothly, while the marginal distribution at $\tau = 0$ is preserved by the transport (Lipman et al., 2023; Liu et al., 2023).

### 2.4. Dynamical Flow

Unlike the compressive flow bridge (3), the dynamical bridge requires a coupling between *pairs* of points in data space and their source representations. Additionally, because we want the dynamical flow $\mathbf{v}_{\theta}$ to exist at every compression level $\tau$, the corresponding linear bridge must involve a *double interpolation* (**Figure 2**). More specifically, given a fixed compression level $\tau \in [0, 1]$, we first interpolate along the compression dimension for each data point as in (3), yielding start and end points $\mathbf{x}_{k\Delta t}^{(\tau)}$ and $\mathbf{x}_{(k+1)\Delta t}^{(\tau)}$. We then perform a second, dynamical interpolation between these points for $s \in [0, 1]$:

$$\mathbf{x}_t^{(\tau)} = (1 - s)\,\mathbf{x}_{k\Delta t}^{(\tau)} + s\,\mathbf{x}_{(k+1)\Delta t}^{(\tau)}. \tag{4}$$

The dynamical flow $\mathbf{v}_{\theta}$ then models the instantaneous dynamical velocity,

$$\mathbf{v}_{\theta} : \ (\mathbf{x}_t^{(\tau)}, \tau, s, \mathbf{x}_{\text{hist}}^{(\tau)}) \ \mapsto \ \partial_t\mathbf{x} \in \mathbb{R}^D.$$

Here, in addition to the bridge variables $\tau$ and $s$, we have allowed a potential dependence on the lag-$h$ history at compression level $\tau$, $\mathbf{x}_{\text{hist}}^{(\tau)} = \left(\mathbf{x}_{(k-h)\Delta t}^{(\tau)}, \ldots, \mathbf{x}_{(k-1)\Delta t}^{(\tau)}, \mathbf{x}_{k\Delta t}^{(\tau)}\right)$ (Zhang et al., 2024).

## 3. Training

Our model comprises both the parameters $(\mathbf{b}, \mathbf{L}, \mathbf{D}, \psi)$ of the encoder (1) and of the compression and dynamical flows $(\phi, \psi)$. The goal is to minimize flow matching losses for both $\mathbf{u}_{\phi}$ and $\mathbf{v}_{\psi}$ while minimizing the dimensionality of the source distribution.

One potential approach to minimizing this latent dimensionality would be to shrink the diagonal scales $\{d_i\}$ using, e.g., standard regularizers (ridge, LASSO) (Hoerl & Kennard, 1970; Tibshirani, 1996) or global-local shrinkage such as the horseshoe (Carvalho et al., 2010). However, we found that even for very severe shrinkage, the generative power of flow matching models is able to compensate nearly down to machine precision, encoding significant information in "unused" dimensions. As a result, such "soft" minimization approaches fail to provide meaningful limits on the dimensionality of source representations. An alternative, which we describe below, is to introduce stochastic sampling over source dimensionality, averaging over the size of this bottleneck during training.

### 3.1. Nested Dropout for Encoder Training

Nested dropout (ND) (Rippel et al., 2014) addresses two key issues previously noted: (1) the need to break permutation invariance among the entries of $\mathbf{D}$ to ensure identifiability; and (2) the requirement of a true low-dimensional source representation. ND addresses these by randomly sampling the rank of $\mathbf{D}$ in a way that enforces an ordering on latent coordinates. More specifically, at each forward pass, it samples a prefix length $K \sim \mathrm{Geom}(p)$, with $\mathbb{E}(K) = 1/p$. This makes $1/p$ an effective latent dimensionality, and marginalizing over $K$ provides control over the capacity of the source representation. In practice, we further truncate $K$ to $[1, D]$, then apply the prefix mask

$$\mathbf{m}_K(i) = \mathbf{1}\{i \le K\}, \qquad \boldsymbol{\mu}_{\boldsymbol{\psi}}^{(K)}(\mathbf{x}) = \mathbf{m}_K \odot \boldsymbol{\mu}_{\boldsymbol{\psi}}(\mathbf{x}), \quad (5)$$

with $\boldsymbol{\mu}_{\boldsymbol{\psi}}^{(K)}$ replacing $\boldsymbol{\mu}_{\boldsymbol{\psi}}$ in (1). As a result, only coordinates $1{:}K$ receive encoder updates in that pass, yielding explicit control of effective dimension.

Note, however, that the flexibility of the nonlinear map $\boldsymbol{\mu}_{\boldsymbol{\psi}}$ that defines the encoder still defines an infinite family of source distributions $p_{\boldsymbol{\psi}, \tau=0}(\mathbf{x})$. Of these, we choose the one that minimizes a masked alignment loss under nested dropout,

$$\mathcal{L}_{\mathrm{align}} = \mathbb{E}_{\mathbf{x}} \mathbb{E}_K \left[ \| \mathbf{x}_t^{(1)} - \mathbf{x}_t^{(0,K)} \|_2^2 \right], \qquad (6)$$

where $\mathbf{x}_t^{(0,K)} = \mathbf{b} + \mathbf{L} \mathbf{D}^{1/2} \boldsymbol{\mu}_{\boldsymbol{\psi}}^{(K)}(\mathbf{x}_t^{(1)})$ is the masked encoder output as in (5).

### 3.2. Flow Matching

We train the compressive flow and dynamical flow using conditional flow matching on the linear interpolations defined in (3) and (4) above.

**Compressive flow matching.** Sample $\tau \sim \mathrm{Unif}[0,1]$ and form $\mathbf{x}_t^{(\tau)}$ via (3). For the linear bridge, the target conditional flow is constant,

$$\mathbf{u}_t^\star = \partial_\tau \mathbf{x}_t^{(\tau)} = \mathbf{x}_t^{(1)} - \mathbf{x}_t^{(0)}.$$

As in (Lipman et al., 2024), we minimize

$$\mathcal{L}_{\mathrm{cf}} = \mathbb{E}_{\mathbf{x}} \mathbb{E}_\tau \left[ \| \mathbf{u}_{\boldsymbol{\phi}}(\mathbf{x}_t^{(\tau)}, \tau) - \mathbf{u}_t^\star \|_2^2 \right]. \qquad (7)$$

**Dynamical flow matching.** At a fixed compression level $\tau$, sample $s \sim \mathrm{Unif}[0,1]$ and construct $\mathbf{x}_t^{(\tau)}$ using the linear within-step bridge (4). For this bridge, the target velocity is constant in $t$,

$$\mathbf{v}_{k,\tau}^\star = \partial_t \mathbf{x}_t^{(\tau)} = \frac{\mathbf{x}_{(k+1)\Delta t}^{(\tau)} - \mathbf{x}_{k\Delta t}^{(\tau)}}{\Delta t},$$

and we minimize

$$\mathcal{L}_{\mathrm{df}} = \mathbb{E}_{\mathbf{x}} \mathbb{E}_{\tau, s} \left[ \| \mathbf{v}_{\boldsymbol{\theta}}(\mathbf{x}_t^{(\tau)}, \tau, s, \mathbf{x}_{\mathrm{hist}}^{(\tau)}) - \mathbf{v}_{k,\tau}^\star \|_2^2 \right]. \qquad (8)$$

**Training objective.** Putting (6), (7), and (8) together, we obtain the combined loss function

$$\mathcal{L} = \alpha \, \mathcal{L}_{\mathrm{cf}} + \beta \, \mathcal{L}_{\mathrm{df}} + \eta \, \mathcal{L}_{\mathrm{align}}. \qquad (9)$$

In all experiments, we set $\alpha = \beta = \eta = 1$. **Appendix A.3** provides a sensitivity check by varying $(\alpha, \beta, \eta)$ on the rotating-ball simulation. We found that the results were robust to these hyperparameters.

## 4. Experiments

A

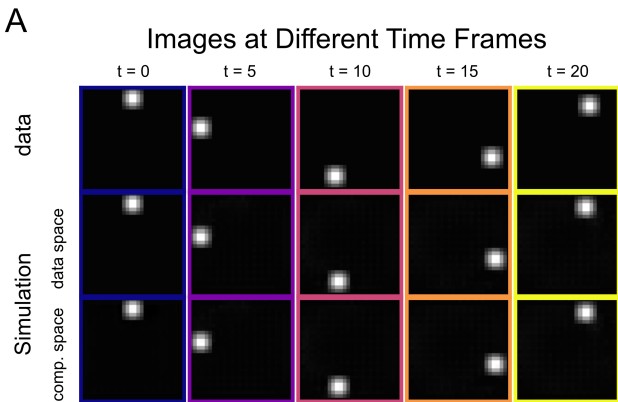

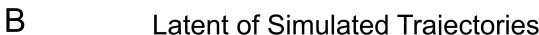

B

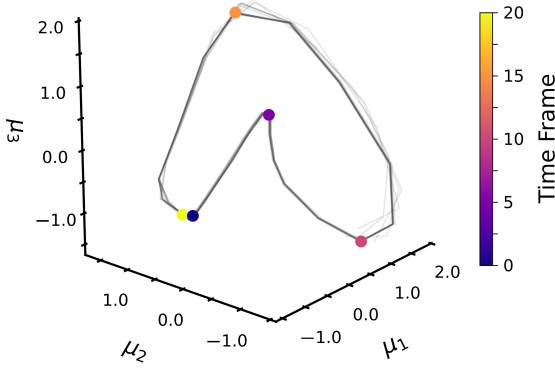

*Figure 3.* **Rotating-ball simulation.** We simulated 10 videos of a ball moving counterclockwise (50 frames, $28 \times 28$) and train DCF with nested dropout ($p = 1/50$) and no history ($h = 0$). (**A**) Ground-truth frames (top) and rollouts decoded in data space (middle, $\tau = 1$), with the corresponding simulated states in compressed space (bottom, $\tau = 0$), shown at $t \in \{0, 5, 10, 15, 20\}$. (**B**) Simulated trajectories visualized in the first three coordinates of the latent space. Colored points correspond to frames in (**A**).

Across experiments, we analyze representations at compression levels $\tau \in \{0, 1\}$. At generation time, we integrate both learned flows (dynamic and compressive) as neural ODEs using an adaptive Dormand–Prince (DOPRI5) solver. We obtain the flowed data representation $\{\tilde{\mathbf{x}}_t^{(\tau)}\}$ (cf. Section 2.3) by two methods: *flowed*, where each observed

frame $\mathbf{x}_t^{(1)}$ is compressed pointwise to level $\tau$ by integrating the compressive flow $\mathbf{u}_\phi$ from $\tau = 1$; and *simulated*, where we first compress an initial frame to $\tilde{\mathbf{x}}_0^{(\tau)}$ and then integrate the dynamical flow $\mathbf{v}_\theta$ forward in time at the same $\tau$. At $\tau = 0$, we map compressed states to encoder-normalized latent coordinates via $\tilde{\boldsymbol{\mu}}_t = (\mathbf{LD}^{1/2})^{-1}\left(\tilde{\mathbf{x}}_t^{(0)} - \mathbf{b}\right)$ and visualize the first three coordinates, which are the most important under nested-dropout ordering. We also project velocities by using $\dot{\tilde{\boldsymbol{\mu}}}_t = (\mathbf{LD}^{1/2})^{-1}\dot{\tilde{\mathbf{x}}}_t^{(0)}$, and plotting dynamical velocities at the start of each step ($s = 0$ in Equation (4)). Experiment train times and parameter choices can be found in **Section B**, cross-run stability checks for identifiability (up to sign) in **Section D**, trajectory roll-out time estimates in **Section C**, and code at our project repository. Supplemental videos, additional experimental details, and raw data for reproducing audio experiments can be found at our project website.

### 4.1. Simulated data

To test the ability of DCF to extract low-dimensional dynamics from high-dimensional data, we simulate 10 short videos of a ball moving counterclockwise (**Figure 3**). Each video comprises 50 time steps, with each frame being a $28 \times 28$ grayscale image. Both the compressive flow $\mathbf{u}_\phi$ and the dynamical flow $\mathbf{v}_\theta$ are parameterized by 4-level convolutional encoder-decoders (U-Net style) with channel widths $\{32, 64, 128, 256\}$ and a 256-dimensional bottleneck embedding. The encoder feature map $\boldsymbol{\mu}_\psi(\mathbf{x})$ is a convolutional VAE with the same multiscale channel schedule. We use no dynamical history. That is, $h = 0$ for $\mathbf{x}_{\text{hist}}^{(\tau)}$ in Section 2.4.

We set nested dropout to $K \sim \text{Geom}(p)$ with $p = 1/50$, so that $K_{\text{target}} = \mathbb{E}(K) = 50$ can provide a generous latent budget. With no additional penalty on $\mathbf{D}$, the fitted scales $\{d_i\}$ decay rapidly (first column of **Figure S1A**). We define the *effective dimension* as the smallest $K$ such that $\sum_{i=1}^{K} d_i \geq 0.95 \sum_{i=1}^{D} d_i$, which yields $K_{\text{eff}} = 30$ in this run. As expected (**Figure 3A**), simulated frames in data space ($\tau = 1$) track the ground-truth ball location over time, while the corresponding rollouts in compressed space ($\tau = 0$, embedded in image space) provide a nearly identical image. Moreover, the simulated latent trajectory in the first three coordinates of $\tilde{\boldsymbol{\mu}}$ forms a smooth closed loop (**Figure 3B**), consistent with the underlying periodic motion. In further experiments imposing additional shrinkage on $\mathbf{D}$ (**Figures S1, S2**), we found that inferred dynamics and latent spaces were *reproducible* across runs. This is expected given that our model produces *identifiable* representations by construction, with additional cross-run stability checks in **Section D**.

For these and subsequent experiments (cf. Sections 4.2,

4.3) we compare DCF against several other competing approaches, namely VAEs (Kingma & Welling, 2014; Rezende et al., 2014), MARBLE (Gosztolai et al., 2025), LFADS (Pandarinath et al., 2018), DMD (Kutz et al., 2016), T-PHATE (Busch et al., 2023), and CEBRA (Schneider et al., 2023). (For larger datasets, we could only train T-PHATE, which needs to load everything into memory, on a subset of data.) Results for these models on the balls dataset are shown in **Figure S4**. Surprisingly, several of these models struggled to identify a simple, low-dimensional manifold underlying the data.

### 4.2. Neural Data

We evaluated DCF on a dataset comprising population neural activity from a center-out reach task performed by non-human primates (592 training trials, 197 held-out test trials; Pei et al., 2021). Each trial's data consisted of smoothed spike counts from 137 neurons over 100 time steps. All methods in Table 1 used the same train/test split and neural time window, with cursor velocity used only afterward for downstream linear decoding. For this experiment, the encoder feature map $\boldsymbol{\mu}_\psi(\mathbf{x})$, the compressive flow $\mathbf{u}_\phi$, and the dynamical flow $\mathbf{v}_\theta$ were all parameterized by multilayer perceptrons (MLPs) with depth 4 and width 256. We use a lag-10 dynamical history ($h = 10$) and nested dropout $p = 1/50$ for all reported results.

As expected, most latent models, including DCF, learned a well-organized latent space in which neural dynamics corresponding to distinct reach targets organized topographically (**Figure S6**). Quantification on a velocity prediction task using these latent spaces (**Table 1**) shows that DCF outperforms other models, despite falling well short of larger, prediction-focused approaches (e.g., Azabou et al., 2023). Additionally, our model achieves higher reconstruction quality of neural activity (i.e., firing rates) on held-out data (**Table S3**). Consistent with latent identifiability up to sign, DCF also recovers stable target-organized 3D latent geometry across five random seeds, whereas LFADS and CEBRA show lower seed-to-seed consistency (**Figure S5**).

### 4.3. Video Data

We next applied DCF to a well-studied behavioral video dataset (Musall et al., 2019), which consists of a single long video with 71,942 frames of size $64 \times 64$. These are challenging data for most dimension reduction methods, since most parts of the frame are highly static, with only intermittent bursts of activity. Here, since we focus on inference, we did not use a train/test split and do not report long-horizon trajectory rollouts. Both flows were parameterized by 4-level convolutional encoder-decoders (U-Net style) with channel widths $\{32, 64, 128, 256\}$ and a 256-dimensional bottleneck embedding. We fit the model with nested dropout

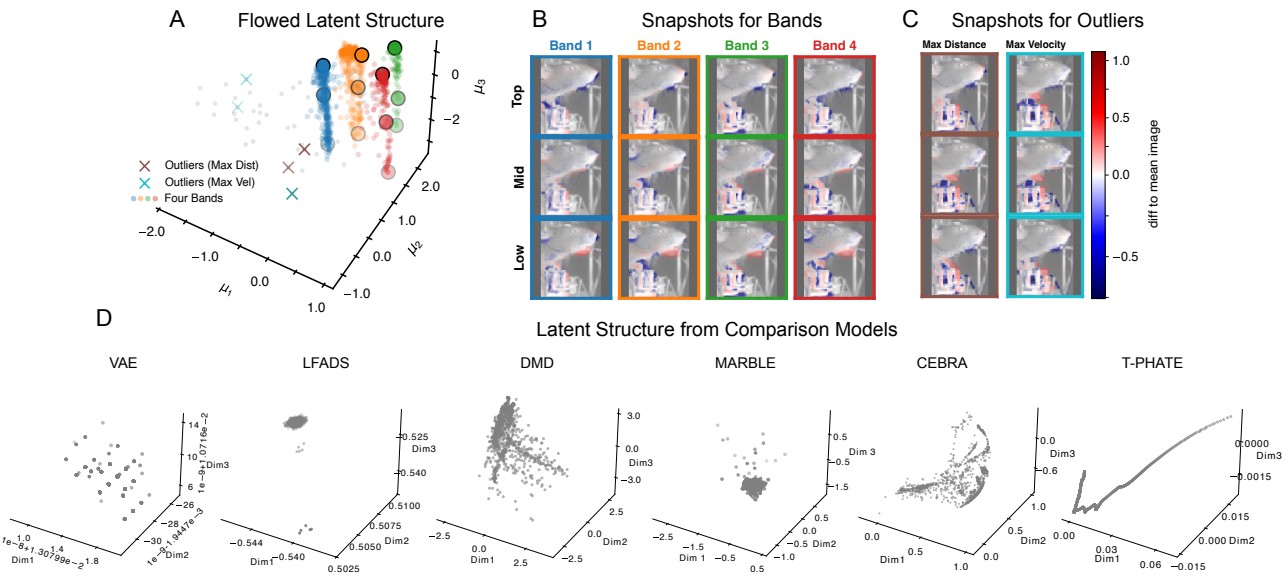

*Figure 4.* **Latent structure in behavioral video.** (**A**) Flowed latent structure forms four prominent bands (colored points), with outliers marked by maximal latent distance (brown ×) and maximal velocity magnitude (cyan ×). (**B**) Representative snapshots along each band (top/mid/low in $\mu_3$). Lowering $\mu_3$ corresponds to stronger mouth movement, paw lift, or both. (**C**) Outlier frames selected by maximal latent distance (left) or maximal velocity magnitude (right), correspond to transient paw and controller movements. (**D**) 3D Latent representations from comparison models. Most either collapse the four bands or fail to capture outliers. See **Figure S7, Figure S8** and **Supplemental Videos at our project website** for additional details on structure identified by our model and comparisons against competing approaches.

| Method | Median $R^2$ ($25^{th}$percentile, $75^{th}$percentile) |
|---|---|
| **DCF (ours)** | **0.304** (0.020, 0.505) |
| VAE* | 0.242 (0.041, 0.427) |
| MARBLE* | 0.195 (−0.083, 0.336) |
| LFADS | 0.311 (0.093, 0.416) |
| DMD* | 0.014 (−0.157, 0.128) |
| T-PHATE* | −0.012 (−0.152, 0.010) |
| CEBRA* | 0.202 (0.010, 0.294) |

*Table 1.* **Model comparisons: neural data.** Decoding (linear predictions) of cursor velocity from 3D latent representations of neural activity. Reported values are $R^2$ quartiles across held-out trials. Asterisks (*) indicate models performing significantly worse than ours (p-value $< 0.05$, one-sided Wilcoxon signed-rank test with Bonferroni correction for multiple comparisons)

$p = 1/50$ and no history ($h = 0$), yielding an effective dimension $K_{\text{eff}} = 30$. Since the behavior is highly repetitive, we visualized only the first 1,438 frames (about $2\%$ of the video), which is sufficient to capture the latent structure.

The latent structure forms four prominent bands in 3D latent space (**Figure 4A**), separated primarily along $(\mu_1, \mu_2)$ while sharing a common within-band axis $\mu_3$. Within each band, lower values of $\mu_3$ correspond to stronger mouth movement, while higher $\mu_3$ is more quiescent (**Figure 4B**).

Across bands, the mean appearance is broadly similar, but each band captures a slightly different baseline visual state, reflected by systematic mean shifts and variability patterns (**Figure S7**). Moreover, outliers selected by maximal latent distance or maximal velocity magnitude (**Figure 4C**) correspond to transient paw and controller movements. Importantly, this latent structure was not well captured by comparison models (**Figures 4D, S8**), and the same four-band structure was recovered across five random seeds (**Figure S5**).

### 4.4. Audio Data

Lastly, we evaluated DCF on a birdsong dataset (262 training trials, 66 test trials) converted to 26 $64 \times 64$ sequential spectrograms. We used the same convolutional architectures as in the video experiment, and the encoder feature map $\boldsymbol{\mu}_{\psi}(\mathbf{x})$ was a convolutional VAE with the same multiscale schedule. We fit the model with nested dropout $p = 1/50$ and no history ($h = 0$), evaluating simulated rollouts.

On training trials, DCF rollouts preserved the syllable-level structure of the motif and reproduced the major time-frequency energy patterns across successive syllables (**Figure 5A**). In the learned 3D latent coordinates, simulated trajectories concentrate on a low-dimensional curved manifold, and transitions between syllables align with regions of larger latent velocity magnitude (**Figure 5B–C**).

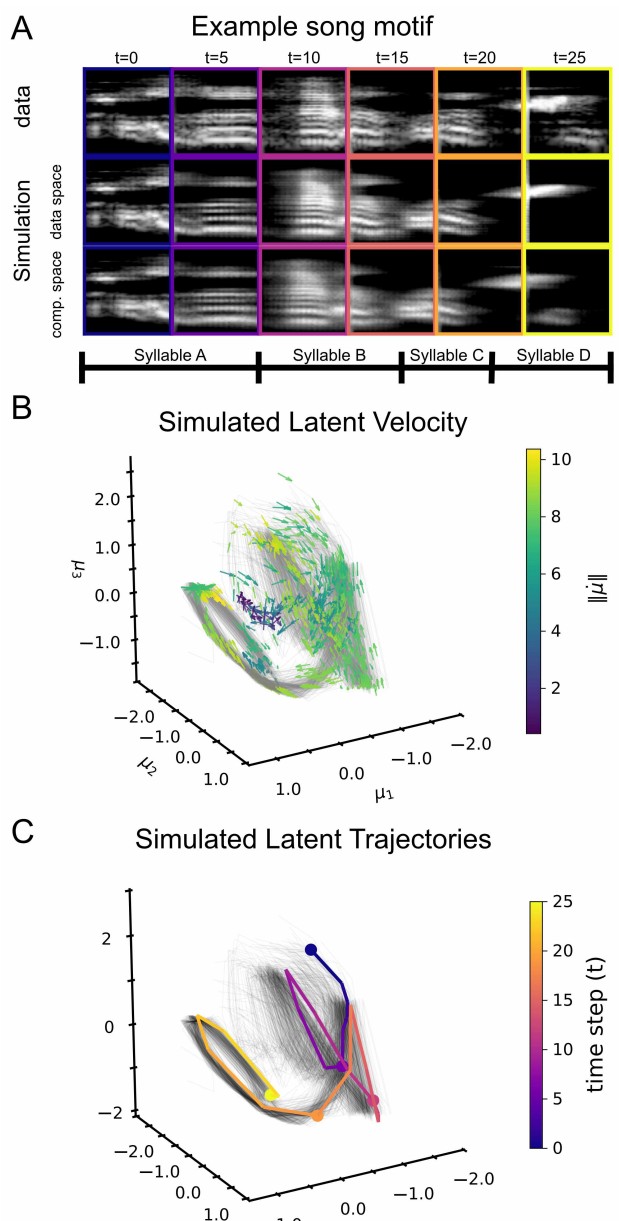

A **Example song motif**

data

Simulation

comp. space  data space

Syllable A    Syllable B    Syllable C    Syllable D

B **Simulated Latent Velocity**

C **Simulated Latent Trajectories**

*Figure 5.* **DCF captures birdsong latent dynamics.** (**A**) Example motif with syllables A–D, shown as ground-truth spectrogram frames (top), *simulated* rollouts decoded in data space (middle, $\tau = 1$), and compressed space (bottom, $\tau = 0$). (**B**) Projected latent velocity field from simulated rollouts in the first three latent coordinates, with arrows normalized to unit length and color indicating velocity magnitude. (**C**) Simulated latent trajectories in the same 3D space, with one representative rollout highlighted and colored by time.

On held-out trials, the same latent geometry is retained and the decoded rollouts remain qualitatively consistent with the ground-truth spectrograms, indicating that the learned dynamics generalize beyond the training set (**Figure S9**). This

is in contrast to typical VAE-based approaches (Goffinet et al., 2021; Sainburg et al., 2020), which preserve structure at the syllable level while failing to smoothly capture dynamics (**Figure S10**).

## 5. Conclusion

Accurate, reproducible methods for understanding and mapping high-dimensional neural dynamics and behavior remain one of the most pressing needs in systems neuroscience. Here, using a flow-matching approach, we combine the strengths of both "mapping" approaches to dimensionality reduction, which assume i.i.d. data, and "dynamical" approaches that focus on the relationships between successive time points. By learning a pair of vector fields, one for compressing data, one for dynamics, we reduce dimensionality while preserving temporal structure. This is made possible by two key innovations: First, rather than specify a simple source distribution like an isotropic Gaussian, we implicitly fit this distribution by learning an encoder (data coupling). Second, we train using Nested Dropout, which allows us to produce an ensemble of true low-dimensional latent spaces. Just as importantly, we have constructed a model that is identifiable up to signs, with the result that our learned latent space is reproducible across training runs.

In experiments, we applied this method to a challenging toy data set, along with population neural data, behavioral video, and audio data. We found that while some comparison models were able to identify latent spaces with known structure in simple cases (neural data), comparison models failed to identify structure in the more challenging behavioral video data, and many failed at even very simple toy examples (balls data). Moreover, the flow-based approach produced both higher-accuracy data reconstructions (**Table S3**) than other generative models and higher latent space decoding accuracy (**Table 1**) than all but LFADS.

**Limitations:** Without smoothing, the modeling framework presented here is not directly applicable to discrete data (e.g., spike counts), which are ubiquitous in neuroscience. However, this limitation can be easily addressed by leveraging recent developments in *discrete* flow-matching techniques (Gat et al., 2024; Campbell et al., 2024), though we leave this for future work. Another limitation lies in the fact that our model does not directly estimate data intrinsic dimensionality but instead requires practitioners to choose the hyperparameter $K_{\text{eff}}$ (cf. Section 3.1), which dictates effective latent space dimensionality. For our experiments, we proposed a simple definition of $K_{\text{eff}}$, which allows the model to train stably and provides a conservative estimate of the true data dimensionality. Future work might further extend our existing approach to allow for latent dimension pruning and adaptive learning of $K_{\text{eff}}$ based on some measure of information preservation in latent space.

## Acknowledgements

This work was supported in part by the U.S. National Institutes of Health (NIH) under grants RF1 DA056376 (JP), F30 MH129086 (DdA), and F31 NS132469 (MM).

## Impact Statement

This work proposes a new deep generative model capable of inferring low-dimensional representations from larger-scale, high-dimensional data while also preserving intrinsic dynamics. Such a model could provide crucial insight into the latent dynamics underlying several neural and biological phenomena of interest, thus having a high potential positive impact in both basic science and medicine. However, advancements in deep generative technologies always have the potential for misuse and can generate content that is misleading, exploitative, or plagiarized. Of note, the model proposed here is *not* aimed at improving generative quality *per se*, but instead seeks to infer more informative and organized latent representations from data. Achieving this goal may prove beneficial in restricting certain types of generation by allowing practitioners to selectively remove, prohibit, or flag regions of the compressed latent space corresponding to potentially harmful content.

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

# A. Additional Results for Ball Video Data

This appendix reports additional ablations for the rotating-ball experiment in Section 4.1.

## A.1. Regularization on D

We study shrinkage regularization on $\mathbf{D} = \mathrm{diag}(d_1, \ldots, d_D)$, which controls the per-coordinate contribution in the compressed endpoint $\mathbf{x}^{(0)} = \mathbf{b} + \mathbf{L}\mathbf{D}^{1/2}\boldsymbol{\mu}_\psi(\mathbf{x})$ when $\mathbf{L}$ is orthonormal and $\boldsymbol{\mu}_\psi$ is normalized. In addition to ridge and LASSO, we consider the horseshoe penalty, a global-local shrinkage prior that strongly suppresses small coordinates while leaving large $d_i$ nearly unchanged due to its heavy tails (Carvalho et al., 2010). Figure S1 summarizes how different penalties reshape the learned scale profile $\{\sqrt{d_i}\}$ while preserving the dominant 3D latent geometry and projected velocity structure.

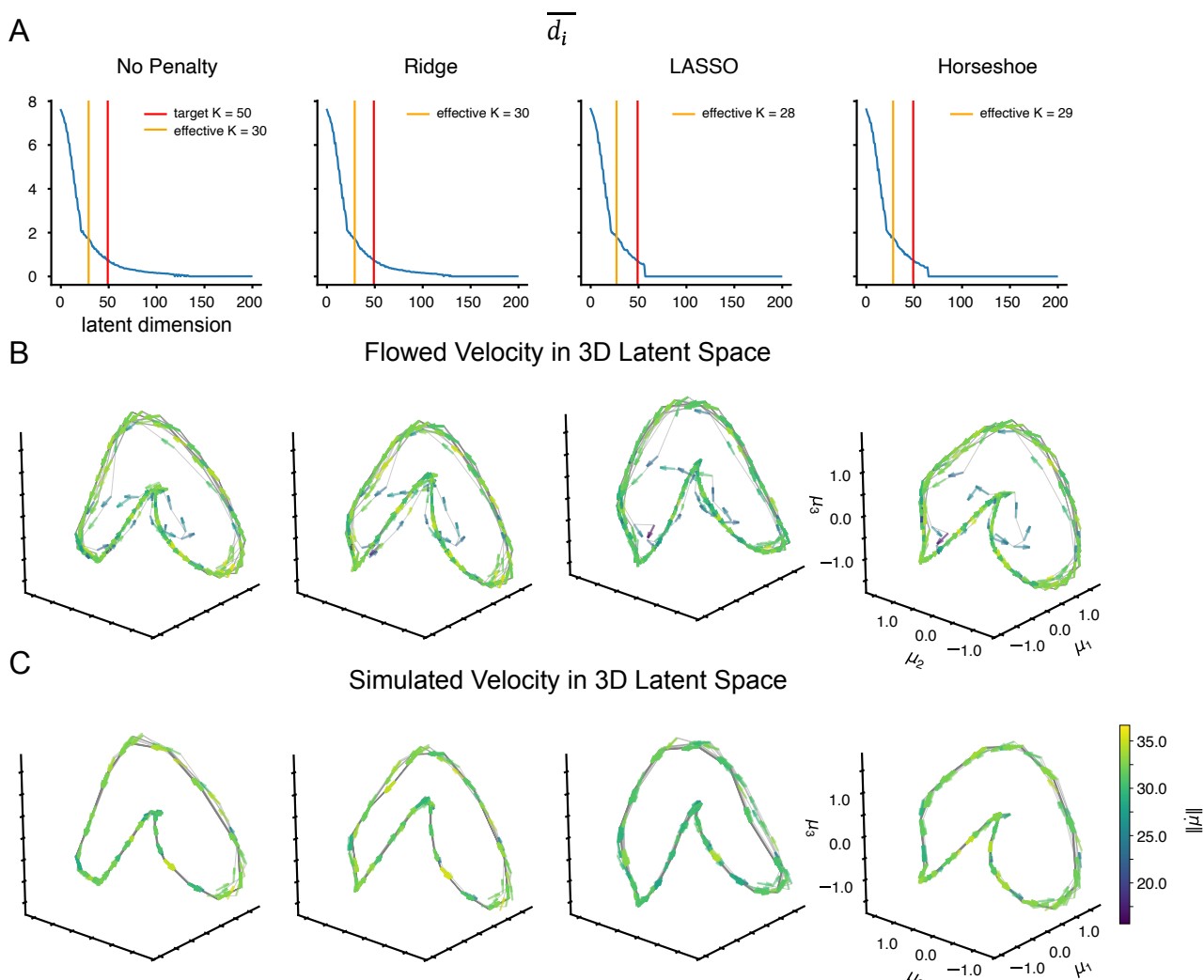

*Figure S1.* **Effect of shrinkage penalties on D for ball simulation.** Same setting as **Figure 3** ($p = 1/50$, $h = 0$), but with different penalties on the diagonal scales $\{d_i\}$: none, ridge, LASSO, and horseshoe (each with a different random seed). (**A**) Learned $\sqrt{d_i}$ as a function of latent index. With orthonormal $\mathbf{L}$ and normalized $\boldsymbol{\mu}_\psi$ (Section 2.2), larger $d_i$ indicates higher contribution of dimension $i$. The red line marks $K_{\text{target}} = \mathbb{E}(K) = 50$, and the orange line marks the effective dimension $K_{\text{eff}}$ (smallest $K$ explaining 95% of $\sum_i d_i$). (**B**) Projected dynamical velocities in the first three latent coordinates for *flowed* trajectories $\tilde{\mathbf{x}}_t^{(0)}$. (**C**) Projected dynamical velocities for *simulated* trajectories (integrating $\tilde{\mathbf{x}}_0^{(0)}$ forward using $\mathbf{v}_\theta^{(0)}$). Arrows are normalized to unit length to emphasize direction, and color indicates the velocity magnitude $\|\dot{\tilde{\mathbf{x}}}_t^{(0)}\|_2$ in the full ambient space. Across penalties, the latent geometry and velocity fields are consistent.

### A.2. Soft 3D latent space

**Figure S2** shows results for experiments using a much tighter nested-dropout budget ($p = 1/3$, so $\mathbb{E}(K) = 3$), showing that the same loop-shaped latent trajectories are retained under a soft 3D representation. Additionally, we compare simulated frames at $\tau = 0$ obtained from either encoder- or flow-based ($\boldsymbol{u}_{\boldsymbol{\phi}}^{(\tau)}$) compression (**Figure S2A**).

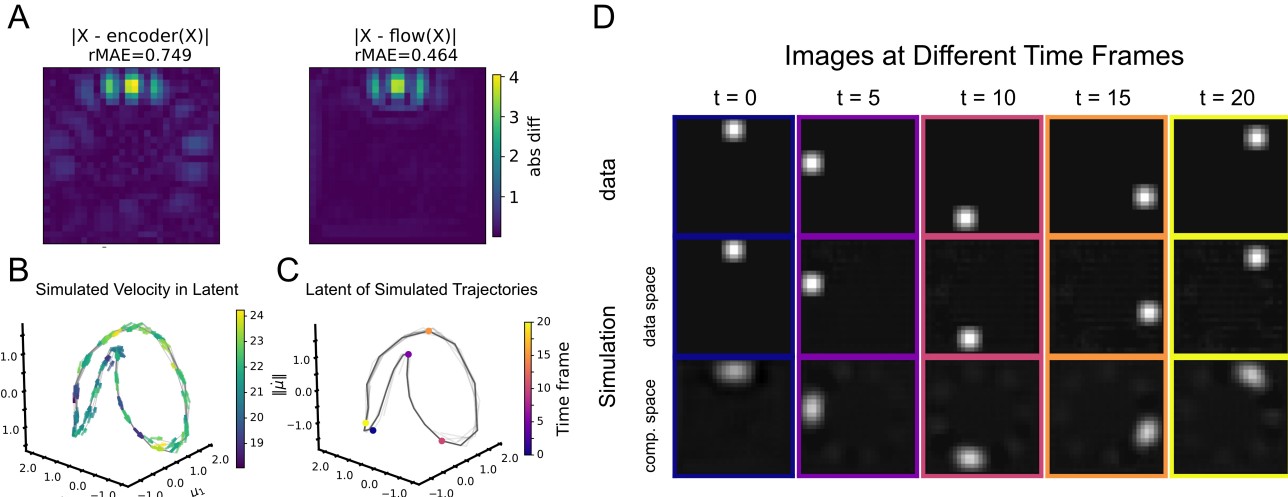

*Figure S2.* **Soft 3D latent representation via nested dropout for ball simulation.** We repeat the rotating-ball experiment with no history ($h = 0$) and a tighter nested-dropout budget $p = 1/3$ (so $K_{\text{target}} = \mathbb{E}(K) = 3$). (**A**) Absolute per-pixel deviation between the original frame and its compression to $\tau = 0$, comparing the encoder endpoint versus the learned compressive flow (rMAE: root mean absolute error per pixel). (**B**) Projected dynamical velocities from *simulated* trajectories in the first three latent coordinates. (**C**) Simulated trajectories in the same 3D latent space, with five representative time points from trial 1 highlighted. (**D**) Corresponding ground-truth frames (top), rollouts in data space (middle, $\tau = 1$), and rollouts in compressed space (bottom, $\tau = 0$).

## A.3. Hyperparameter settings for loss terms

In the main text, we use $\alpha = \beta = \eta = 1$ for all experiments. Here, we show that the results, including latent geometry, are robust under different loss weights, i.e., $(\alpha, \beta, \eta) = (5, 1, 1), (1, 5, 1),$ and $(1, 1, 5)$.

We study sensitivity to the loss weights in (9). Using the same rotating-ball setting as Section 4.1 ($p = 1/50$, $h = 0$ and no penalty of $\mathbf{D}$), we retrain DCF while increasing one weight by $5\times$ and keeping the other two fixed: $(\alpha, \beta, \eta) \in \{(5, 1, 1), (1, 5, 1), (1, 1, 5)\}$. Figure S3 compares the learned scale profile ($\sqrt{d_i}$) and the projected 3D latent velocities for both flowed and simulated trajectories.

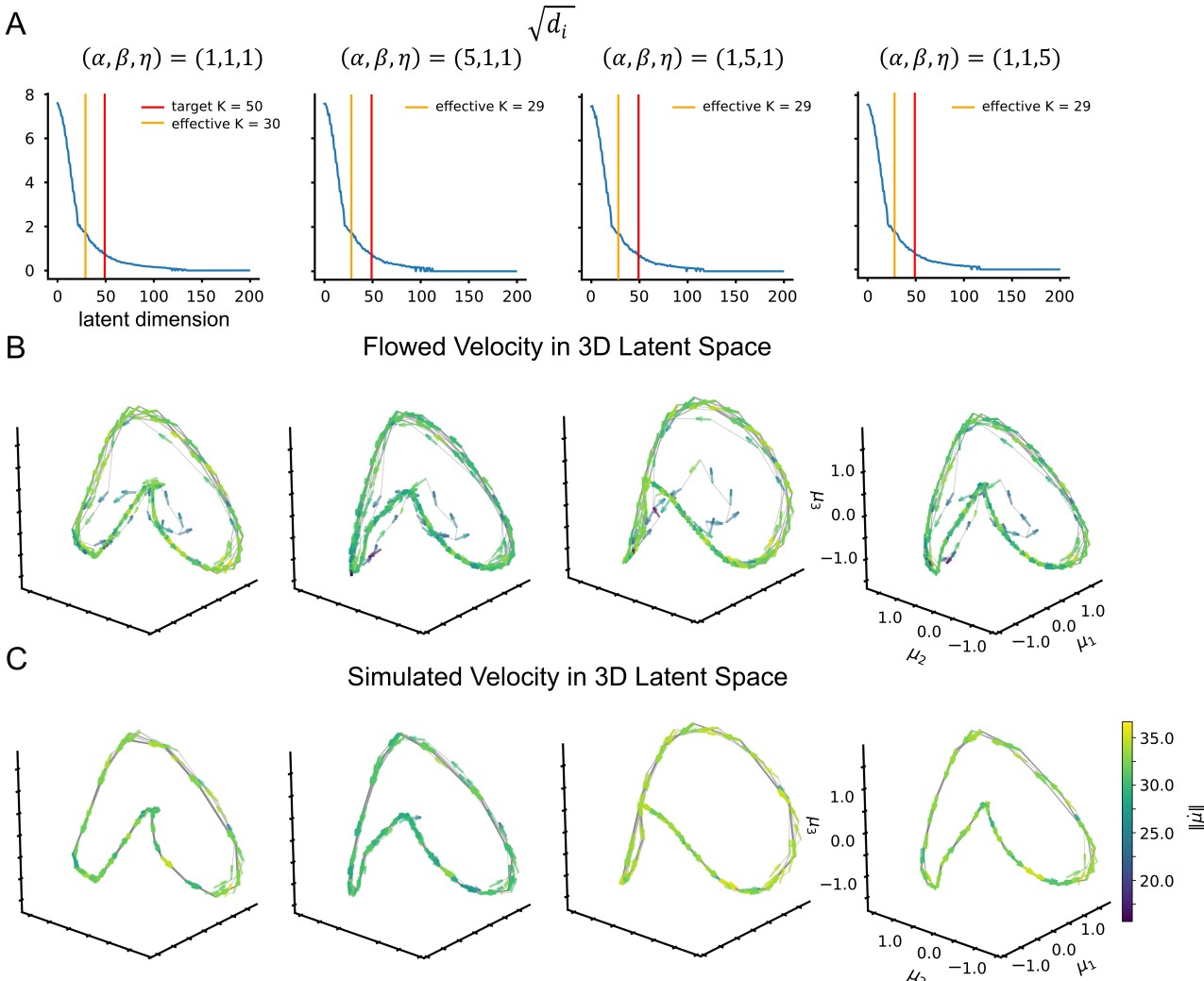

*Figure S3.* **Sensitivity to loss weights (rotating-ball simulation).** Same setting as Section 4.1 ($p = 1/50$, $h = 0$ and no penalty on $\mathbf{D}$), retrained with $(\alpha, \beta, \eta) \in \{(1, 1, 1), (5, 1, 1), (1, 5, 1), (1, 1, 5)\}$. **(A)** Learned scale profile $\{\sqrt{d_i}\}$ versus latent index. The red line marks $K_{\text{target}} = \mathbb{E}(K) = 50$, and the orange line marks the effective dimension $K_{\text{eff}}$ (smallest $K$ explaining 95% of $\sum_i d_i$). **(B)** Projected dynamical velocities for *flowed* trajectories in the first three latent coordinates. **(C)** Projected dynamical velocities for *simulated* trajectories. Arrows are normalized to unit length to emphasize direction, and color indicates velocity magnitude.

## A.4. Latent spaces for comparison models on the balls toy dataset

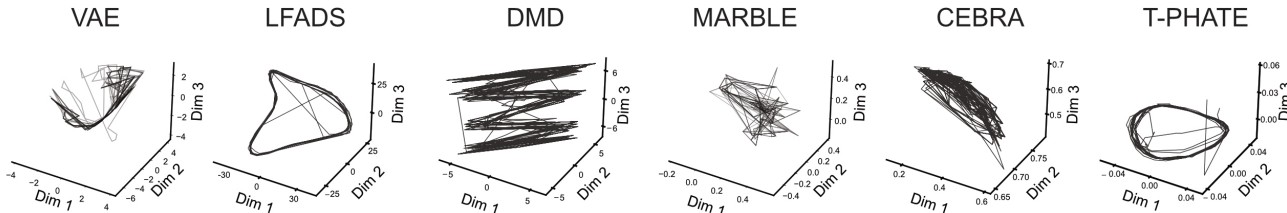

*Figure S4.* **3D latent representations of comparison models on the rotating ball dataset** Nearly all comparison models fail to capture the cyclical latent structure of the ball dataset; those that do display more variable latent trajectories than DCF.

# B. Training Time and Experiment Parameters

| Dataset | $K_{\text{target}}$ | Lag | Penalty | DCF Train Time (Mimgs) | DCF Train Time (hours) | LFADS Train Time (hours) |
|---------|---------|-----|---------|------------------------|------------------------|--------------------------|
| Ball | 50 | 0 | None | 4.27 | 1.97 | 0.045 |
| | 50 | 0 | Ridge | 4.27 | 1.97 | – |
| | 50 | 0 | Lasso | 4.27 | 1.97 | – |
| | 50 | 0 | Horseshoe | 4.27 | 1.97 | – |
| | 3 | 0 | None | 4.27 | 1.17 | – |
| Maze | 50 | 10 | None | 55.70 | 8.51 | 0.066 |
| Mouse | 50 | 0 | None | 20.82 | 5.70 | 0.251 |
| Bird | 50 | 0 | None | 32.61 | 8.15 | – |

*Table S1.* **Experiment training times and parameters.** Total DCF training times, measured in both $10^6$ images (i.e., Mimgs) and hours across our different simulated and neuroscience data experiments. LFADS training times are also reported for datasets where the comparison was run. All experiments were conducted on single NVIDIA RTX 4090 GPUs, with per-GPU batch size of 16. $K_{\text{target}}$ specifies the geometric distribution used with nested dropout, and lag measures the size of the history window used for contextual prediction (i.e., $\mathbf{x}_{\text{hist}}^{(\tau)}$).

# C. Trajectory roll-out times

| Dataset | K | Lag | Total Trials | Trial Length | Total Roll-Out Time ($\tau = 0$) | Total Roll-Out Time ($\tau = 1$) |
|---------|---|-----|--------------|--------------|----------------------------------|----------------------------------|
| Ball | 50 | 0 | 10 | 50 | 2s | 2s |
| | 3 | 0 | 10 | 50 | 2s | 2s |
| Maze | 50 | 10 | 592 | 100 | 4s | 5s |
| Mouse | 50 | 0 | 1 | 1438 | 1.45min | 2min |
| Bird | 50 | 0 | 262 | 26 | 13s | 13s |

*Table S2.* **Trajectory Roll-Out Times.** Roll-out times at $\tau = 0$ and $\tau = 1$ for the different experiments performed. Roll-out simulations were conducted using single NVIDIA RTX 4090 GPUs, and trials were stacked. All values included here are for experiments without added penalties on $\mathbf{D}$. Use of additional regularizers on $\mathbf{D}$ yields comparable roll-out times.

# D. Cross-run stability

To empirically verify that our latent coordinates are indeed identifiable up to sign, we evaluated cross-run stability of the learned 3D latent representations using independent random seeds. For each quantitative comparison, we aligned

coordinate signs across runs and computed the mean cosine similarity between the resulting 3D latent representations. For the rotating-ball experiment, stability is already shown in Figures S1 and S2, where the latent loop and projected velocity field are preserved across random seeds, shrinkage penalties, and latent-budget choices.

For the Musall mouse video experiment, DCF recovered the same 3D latent geometry across five random seeds, and $k$-means identified the same four clusters as in Figure 4. For the monkey center-out reach experiment, DCF also recovered stable target-organized trajectories across seeds. In contrast, LFADS and CEBRA showed substantially lower cross-seed consistency on the monkey reach data.

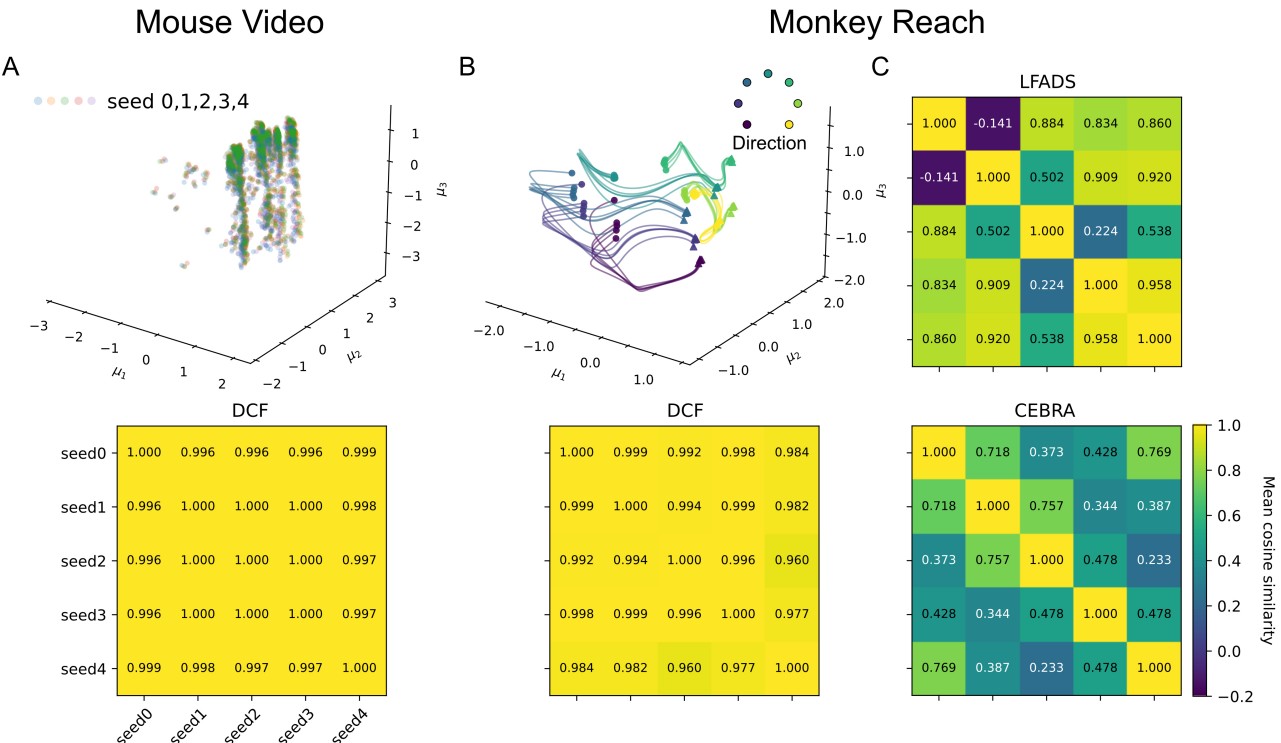

*Figure S5.* **Cross-run stability of learned latent representations.** (**A**) Musall mouse-video experiment. Top: sign-aligned 3D DCF latent representations from five random seeds. Bottom: pairwise mean cosine similarity between seeds. (**B**) Monkey center-out reach experiment. Top: sign-aligned, target-averaged 3D DCF latent trajectories across five random seeds. Bottom: pairwise mean cosine similarity between seeds. (**C**) Pairwise mean cosine similarity for LFADS and CEBRA on the monkey reach experiment. All heatmaps use the same color scale.

# E. Additional Experiments for Monkey Center-Out Reach (Maze) Data

This appendix reports results obtained on neural center-reach out (maze) data. **Table S3** shows that our model easily outperforms all other competing (generative) approaches in reconstructing firing rates on held-out (test) data. **Figure S6** shows that our model correctly infers topographically organized trajectories corresponding to the different (center-reach) directions used throughout the experiment. Flowed trajectories for our model were obtained by compressing the observed data $\mathbf{x}_t^{(1)}$ pointwise to latent space ($\tau = 0$) by integrating the compressive flow $\mathbf{u}_\phi$ at each $t$.

| Method | Median $R^2$ ($25^{th}$percentile, $75^{th}$percentile) |
|---|---|
| **DCF (ours)** | $0.999\ (0.998, 0.999)$ |
| VAE | $0.618\ (0.559, 0.678)$ |
| LFADS | $0.595\ (0.527, 0.661)$ |

*Table S3.* **Model comparisons: neural data.** Reconstruction of neural activity (firing rates) on held out test-data. Reported values are $R^2$ quartiles across held-out trials.

## Latent Trajectories Colored by Target Location

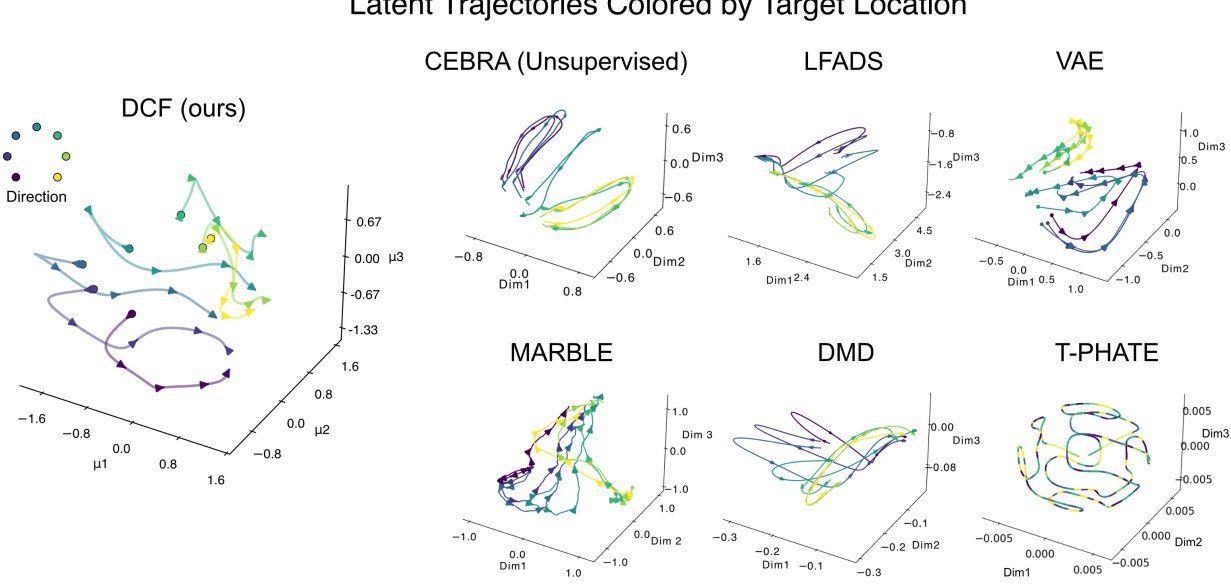

*Figure S6.* **3D latent representations of neural data. Left:** Flowed mean trajectories (compressing data pointwise via compressive flow $\mathbf{u}_\phi$) in the first three latent coordinates. Color indicates monkey reach direction; lines are averaged across reach direction. **Right:** 3D latent trajectories of comparison models. The latent representation of CEBRA (without supervision), LFADS, VAE, MARBLE and DMD are averaged across monkey reach location. T-PHATE latent representation is unstructued and therefore not averaged.

## F. Additional experiments on mouse video data

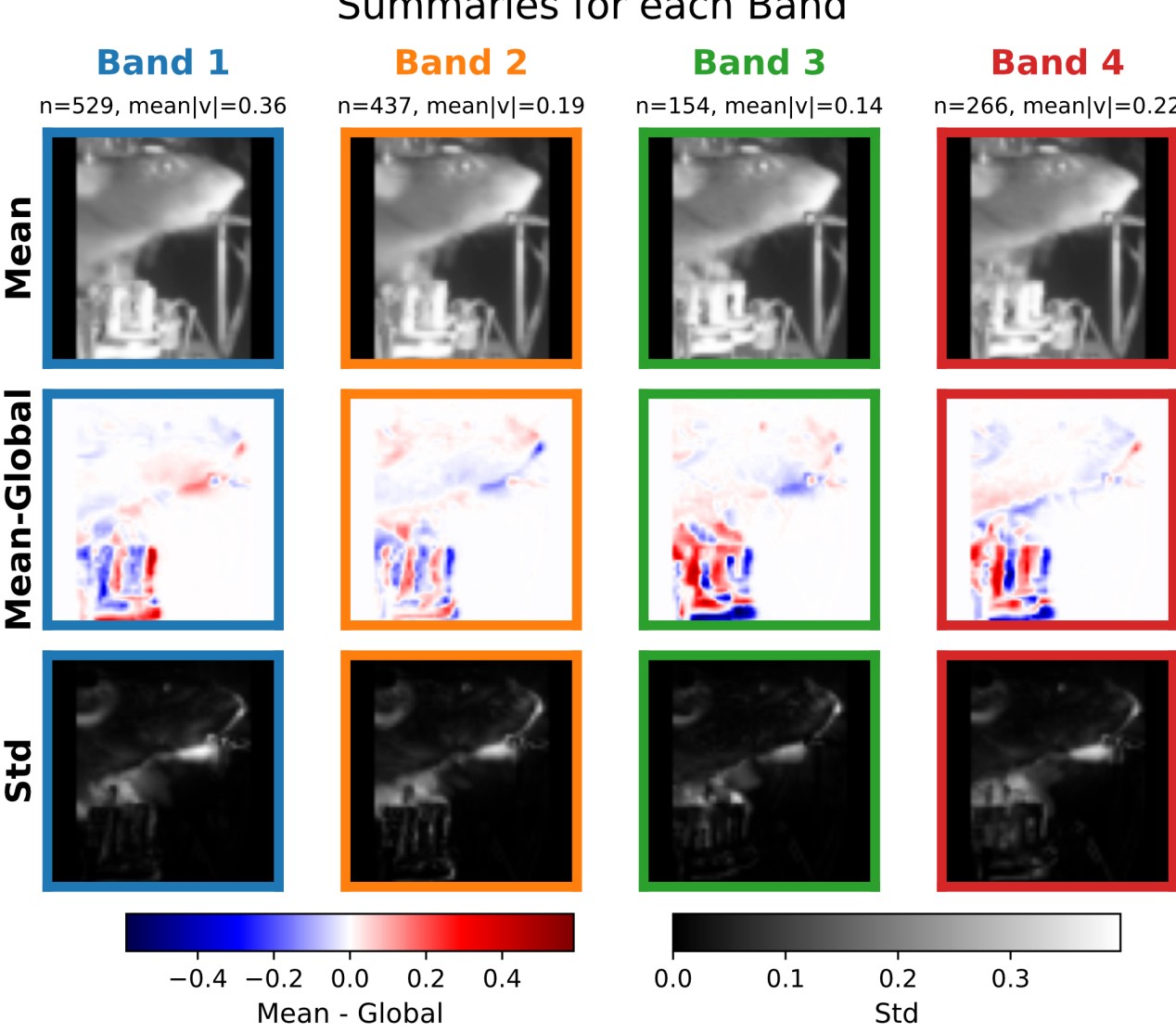

*Figure S7.* **Band-level summaries for mouse video data.** For each band in **Figure 4A**, we show the mean frame (top), the mean deviation from the global mean (middle, mean minus global), and the per-pixel standard deviation (bottom). We report the number of frames $n$ and the mean velocity magnitude within each band. Of note, moving top to bottom across bands changes video from periods of small, more quiescent activity to periods of larger paw and mouth/tongue movement. In general, our blue band captures mostly wider mouth and tongue movements (i.e., repetitive licking), whereas our orange, red, and green bands capture a mix of both paw and mouth/tongue movement, with movements becoming more subtle as we move from orange → red → green. Please check our project website which contains a supplemental video (`Latent_Plot_DCF.mp4`) showcasing these frame transitions as we move in our inferred latent space.

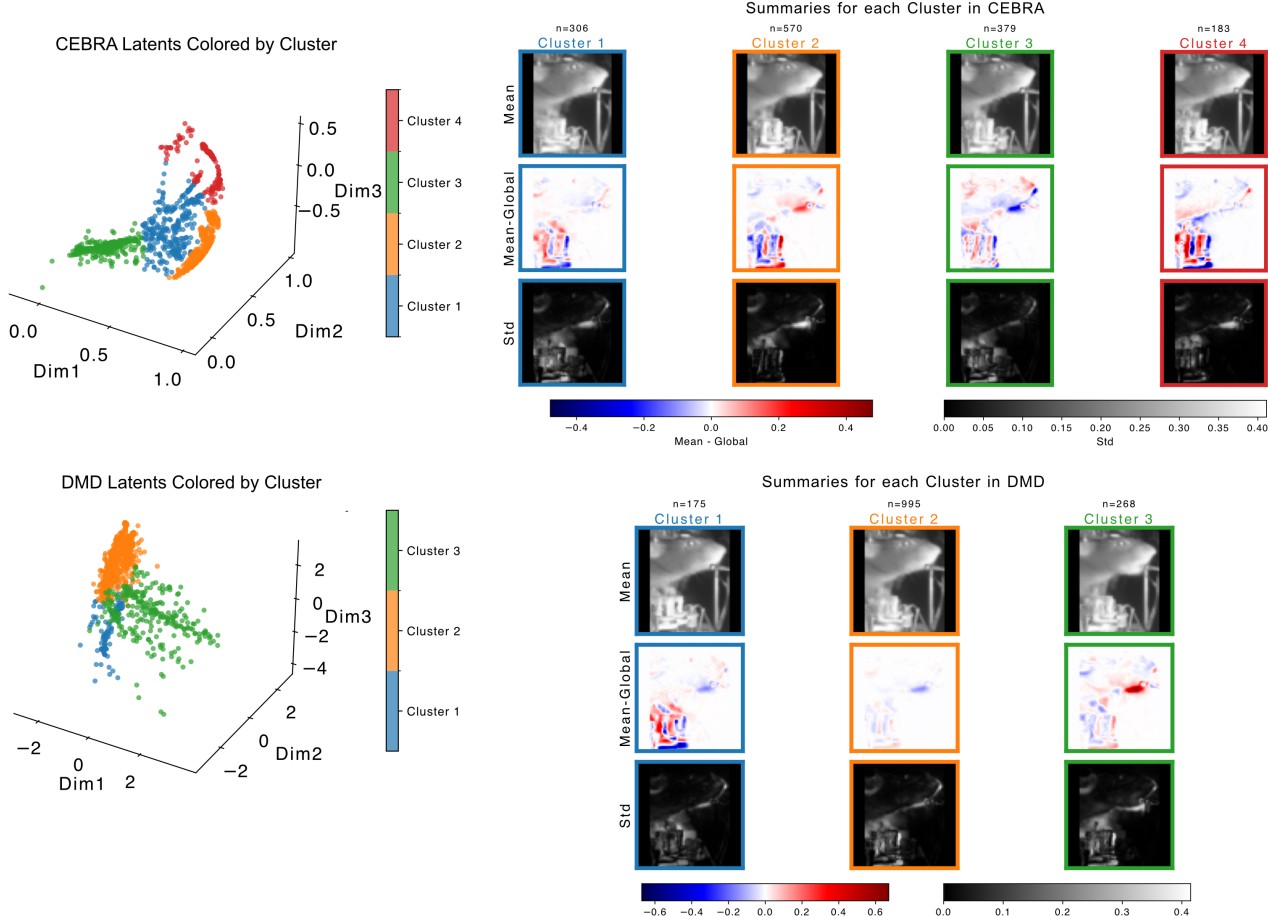

*Figure S8.* **Latent space structure in DMD and CEBRA comparison models.** Latent spaces (left columns, color-coded by clusters) and cluster mean frame, mean deviation from global mean, and per pixel standard deviation (right columns, same construction as in **Figure S7**). In general, both DMD and CEBRA fail to separate outliers related to brisk controller movements from rest of data. Additionally, clusters identified often mix periods of quiescent activity with movement and mix more than one type of movement within the same cluster, providing overall less structured latent spaces. Please check our project website which contains supplemental videos `Latent_Plot_DMD.mp4` and `Latent_Plot_CEBRA.mp4` showing frame transitions as we move through the latent spaces inferred by these competing models.

## G. Additional experiments on bird audio data

Dataset (raw audio and labeled syllables) and accompanying pre-processing code (`get_specs.py`) for bird audio experiments are included in our project website under the `Resources` tab. Raw audio data are included as `.wav` files and are named by date of data collection (YYYYMMDD format). Syllable labels are included as `.txt` files, with similar date-name structure.

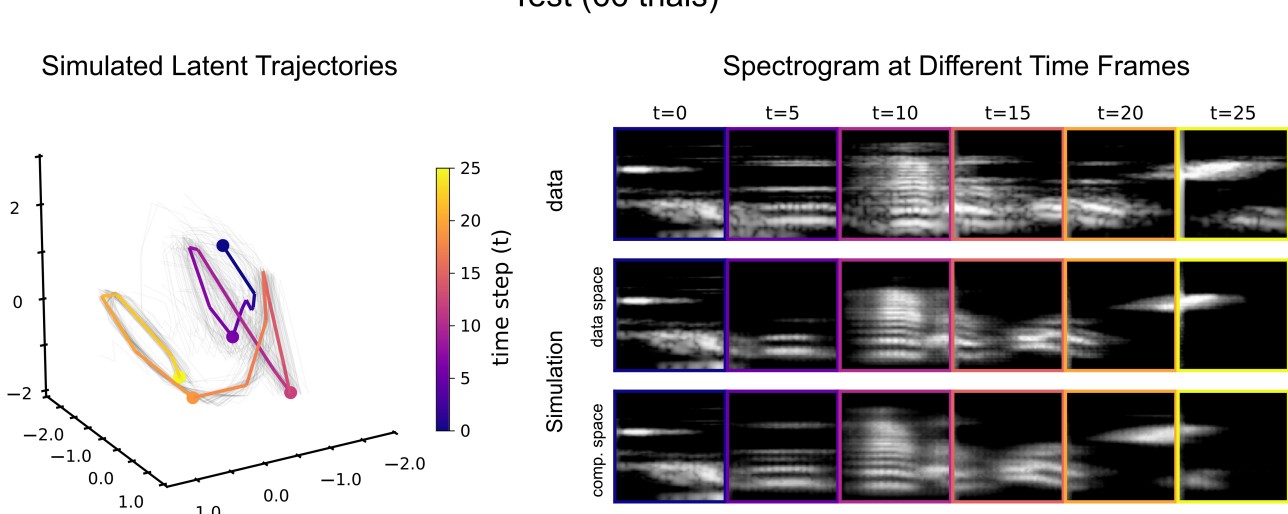

*Figure S9.* **Birdsong rollouts on the test set.** Left: simulated latent trajectories in the first three coordinates of $\tilde{\mu}$ for held-out trials. Right: ground-truth spectrogram frames (top) and corresponding simulated rollouts decoded in data space (middle, $\tau = 1$) and compressed space (bottom, $\tau = 0$), shown at matched time points.

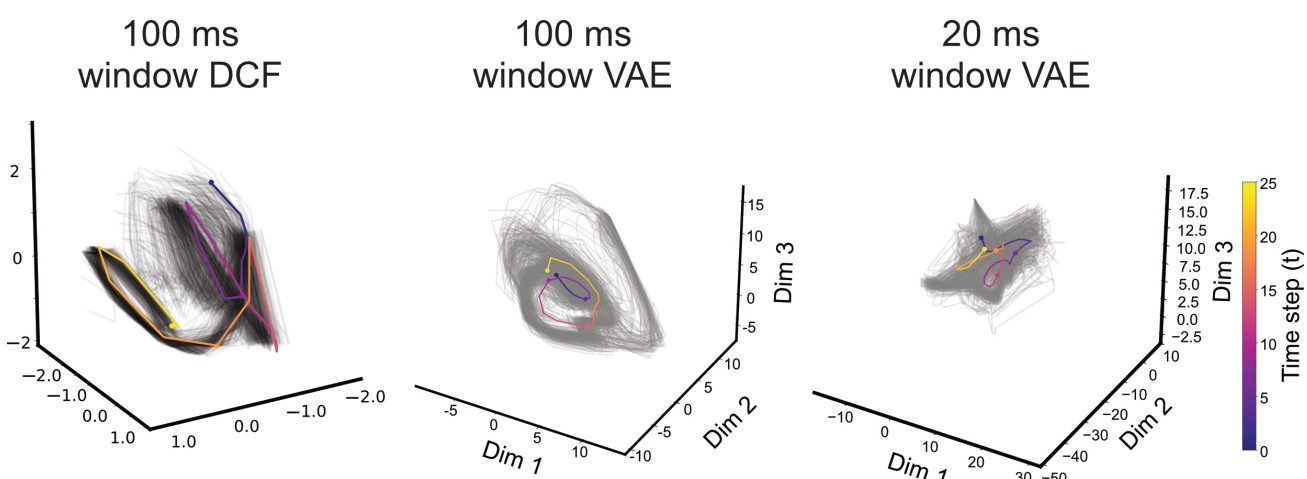

*Figure S10.* **Comparison embeddings using a 3d VAE.** Left: simulated trajectories from a DCF model (cp. **Figure 5**C). Middle: embeddings of a VAE trained using 100 ms long spectrogram windows. Right: embeddings of a VAE trained using 20ms long spectrogram windows. In general, VAEs with short data windows produce latent spaces with disorganized temporal structure, while longer data windows exhibit more obvious structure but more variability than DCF.

