# OpenReview forum: "Dynamic Compression Flows for Neuroscience Data"
_ICML.cc/2026/Conference — ICML 2026 regular_

### Official Review · Reviewer_mcAW · 2026-03-08

**Soundness:** 2
**Presentation:** 3
**Significance:** 2
**Originality:** 2
**Overall Recommendation:** 3
**Confidence:** 3

**Summary:**

The paper presents a flow match framework for high-dimensional neural dynamics that jointly learns the projection to a low-dimensional latent space and the temporal evolution within it. The model uses a hierarchical bottleneck (nested dropout) to learn an identifiable, multi-scale representation. The authors demonstrate recovery of latent structure on synthetic, neural, and behavioral data, and outperform standard benchmarks.

**Compliance With Llm Reviewing Policy:**

Affirmed.

**Final Justification:**

The paper presents an interesting dual flow-matching framework for neural data. The rebuttal provided useful new evidence, particularly the cross-seed cosine similarity analysis, which partially addresses some of my concerns at the empirical level. However, the identifiability argument is tied to the encoder head, but the compressive flow can rearrange latent geometry at $\tau=0$, so encoder identifiability alone does not guarantee that the recovered latent structure is unique. Hence, I maintain my score of 3.

**Key Questions For Authors:**

- The authors critique other models for assuming `simple noise forms`, yet you utilize an $L_2$ penalty. Since $L_2$ fundamentally assumes a Gaussian distribution, how is this model less restrictive in its noise assumptions than some of the methods you cite?
- How does the model handle the potential for equivalent solutions regarding rotations and transformations in the latent space? Is the solution invariant to these, and if so, how do you ensure a unique or consistent structure is recovered?
- I believe that the optimization problem is not convex. If not, how robust is the recovered structure to different initializations?
- How do you determine the appropriate $\Delta t$ for the ODE solver, and how sensitive is the latent recovery to this value?
- In the introduction, you discuss dynamics repeatability as a limitation of current methods. While this is indeed a limitation of some models (e.g., SLDS), it does not apply to all existing methods (e.g., DLDS, LFADS). Can you clarify the claim that existing models struggle with non-repeatability, or be more precise about the specific advantages of your approach compared to these methods?

**Limitations:**

yes

**Strengths And Weaknesses:**

## Strengths

- The paper addresses a significant challenge in latent variable modeling by moving away from flat representations and instead proposing a method to identify multi-scale hierarchical structures in neural and behavioral data.

- The application of flow-matching to jointly learn the mapping between data and latents alongside the temporal evolution is an interesting approach that could simplify the training of complex dynamical systems.

- The authors demonstrate the method across diverse datasets, showing better recovery of latent geometry than standard benchmarks.

- The paper is overall well written.

## Weaknesses

- I am concerned about a hidden mathematical assumption regarding the high-dimensional data that is not fully addressed. In neuroscience, high-dimensional data is often extremely jagged and noisy. The model, from what I understand, implicitly relies on defining a flow field directly in the high-dimensional observation space, which implicitly assumes that the data itself exhibits smooth and differentiable transitions. However, if the mapping from latent states to observations is noisy or "tangled" enough, a coherent flow field may not actually exist at the data level. This forces the model to artificially impose smooth dynamics. I would have liked to see a discussion on whether this assumption is realistic for messy neural recordings or how the model behaves when these high-dimensional transitions are inherently non-smooth, or at least more explicitly stating what the math assumptions about the data are for the model to work.

- I have reservations regarding the recovery of the latent structure and identifiability. I can imagine multiple low-dimensional solutions that are effectively equivalent; specifically, I think the space is invariant to certain rotations and transformations. It is unclear how the model copes with this or ensures a unique solution. Furthermore, since the problem is likely non-convex, I am concerned that the recovered structure might just be a local minimum dependent on the arbitrary choice of non-linearity or the $\Delta t$ used in the ODE solver.

- The presentation in certain sections is confusing. For instance, the paragraph in lines 160-164 (right column) feels misplaced. QR decomposition is mentioned earlier just as an example, but then treated here as the actual method for decomposing $L$. Point (2) in that same paragraph is discussed as an advantage before it is even explained (only later in Section 3.1), which makes the rationale hard to follow. I think rephrasing that paragraph to very briefly describe these concepts instead of referring to them as things that should be known, or alternatively, relocate this paragraph to after these ideas are introduced, would be helpful.

- The visualizations in Figure 4 are not clear enough and could demonstrate the claims better. In Figure 4b, the bands look so similar that this panel feels like a misuse of space and it is very hard to understand what the bands' meaning is based on that illustration. I see that in the appendix (fig S7) the authors dive more into the differences, and I think that these appendix figures showing the differences are a better use for the main text. However, I still wonder if the differences between these bands are even statistically significant or if similar results could be obtained by randomly sampling snapshots of the data. An idea is to color all baseline models in Figure 4D using the same colors as the clusters in Figure 4A, which would be a much more effective way to show if these bands are actually being captured similarly across models.

- Another issue is the claims regarding noise compared to other methods. The authors claim that most models are limited by assuming simple noise forms, yet their method relies on an $\ell_2$ penalty, which fundamentally assumes a Gaussian distribution as well.

### Smaller:

- I believe that on line 190, $\psi$ should likely be $\theta$?

- The authors claim in line 080 that other models struggle with non-repeatability--that is correct for some methods (e.g. SLDS and other switching based approaches) but there are a lot of existing methods that do not require repeatability of the signal (e.g., [1,2]).


*[1] Mudrik, N., Chen, Y., Yezerets, E., Rozell, C. J., & Charles, A. S. (2024). Decomposed linear dynamical systems (dlds) for learning the latent components of neural dynamics. Journal of Machine Learning Research.*

*[2] Pandarinath, C., O’Shea, D. J., Collins, J., Jozefowicz, R., Stavisky, S. D., Kao, J. C., ... & Sussillo, D. (2018). Inferring single-trial neural population dynamics using sequential auto-encoders. Nature methods*

---

> ### Author Rebuttal · Authors · 2026-03-30
>
> We thank the reviewer for their thoughtful comments. In answer to the reviewer's questions:
>
> > I am concerned about a hidden mathematical assumption regarding the high-dimensional data that is not fully addressed. ...
>
> The reviewer raises an important concern. However, our aproach _does not_ assume the transition from source to target distributions is smooth.
> Rather, the model assumes a smoothly varying flow field/score function, as in standard diffusion- and flow-based modeling. More specifically, in flow matching, we assume smooth dynamics of the _marginal distributions_ of data, _not the sample paths_. That is, individual data trajectories may be jagged and noisy while the distributions of data change smoothly.
>
> This assumption is what allows diffusion- and flow-based generative models to handle complex, high-dimensional data directly in the **original data space**, as has been demonstrated for a wide variety of data types in the literature, such as [1] [2].
>
> > I have reservations regarding the recovery of the latent structure and identifiability....
>
> To clarify: Our claim is not about identifiability of the flow network _parameterization,_ which involves a non-convex optimization. Rather, our claim is that the encoder head $\mu_\psi$ is identifiable in our chosen parameterization. See our reply to Reviewer 2Pev for a more detailed argument on this identifiability claim.
>
> > The presentation in certain sections is confusing. ...
>
> Thanks for pointing this out. We will revise this part for clarity. There are multiple ways to construct an orthonormal $L$, and in our implementation, we use non-pivoted QR.
>
> > The visualizations in Figure 4 are not clear enough and could demonstrate the claims better.
>
> Thank you for the suggestion. We agree Figure 4 can be improved, and we will revise it to make the relevant comparisons and claimed differences clearer. We also would like to point reviewers to animations we included as part of our supplemental materials, which showcase how dynamics progression in the learned latent spaces relate to specific changes in the video frames for both our model (DCF) and competing approaches. We believe these videos provide a better visualization of our claims. Please also refer to our answer to Reviewer 2Pev regarding cluster consistency.
>
> > The authors critique other models for assuming simple noise forms, yet you utilize an L_2 penalty.
>
> The $L_2$ terms in our objective are regression losses used solely for the purpose of learning the vector fields as part flow matching. These do not assume an explicit Gaussian observation model for the data. In flow matching, the _conditional_ Gaussian bridge is an auxiliary construction used to learn the transport from source to target in an efficient manner. The _marginal data distribution_ is, in fact, arbitrary, rather than Gaussian, as claimed and proved in the flow matching literature [1] [2].
>
> > How do you determine the appropriate $\Delta t$ for ODE solver, and how sensitive is the latent recovery to this value?
>
> In the inference/ data generation stage, we use an adaptive DOPRI5 solver for ODE integration, so there is no $\Delta t$ to tune. In the training stage, the training is "simulation-free" (i.e., no numerical integration is needed because the conditional flow is exactly solvable), so the choice of ODE solver does not influence model training.
>
> > Can you clarify the claim that existing models struggle with non-repeatability, or be more precise about the specific advantages of your approach compared to these methods?
>
> We apologize for the confusion. Our claim is not that all existing methods fail under non-repeatable dynamics, but some models make assumptions that may be brittle in these cases. For example, models that assume neural dynamics follow a (non)linear dynamical system with small observation noise likewise assume a flow field $\boldsymbol{\psi}(\mathbf{x})$, but this assumption is on the sample paths, which as the reviewer notes above, may be highly noisy and jagged. Flow matching, by contrast, is a model of the _distributional flow_ of data and so need not assume that individual trajectories always follow the same (mean) path ($\mathbf{\dot{x}} = \mathbf{f(x)} + \boldsymbol{\varepsilon}$) given the same latent state. In this way, our model accommodates the complexity of neural data without making strong assumptions about individual trajectories.
>
> [1] Lipman, Y., Chen, R. T. Q., Ben-Hamu, H., Nickel, M., & Le, M. (2023). Flow Matching for Generative Modeling. In International Conference on Learning Representations (ICLR 2023).
>
> [2] Tong, A., Fatras, K., Malkin, N., Huguet, G., Zhang, Y., Rector-Brooks, J., Wolf, G., & Bengio, Y. Improving and Generalizing Flow-Based Generative Models with Minibatch Optimal Transport. Transactions on Machine Learning Research (TMLR), 2024.

---

> > ### Author Rebuttal · Reviewer_mcAW · 2026-04-01
> >
> > I appreciate the authors' responses. However, two of my original concerns remain partially unresolved.
> > 1) Regarding my weakness on identifiability and non-convexity: the authors narrow the identifiability claim to the encoder head and defer to Reviewer 2Pev, but my concern was about whether the recovered latent structure is robust to different initializations given the non-convex optimization. The flow can rearrange geometry at τ=0, so consistency of the encoder head alone does not resolve this. Demonstrating agreement across random seeds on a dataset beyond the ball simulation would help.
> > 2) Regarding my weakness on the L2/noise assumption: I accept that the L2 in flow matching is a training objective for learning a vector field, not a Gaussian observation model in the traditional sense. However, the paper motivates itself by failures under noise-dominated and heavy-tailed regimes. Even if the data distribution is theoretically unconstrained, the L2 training signal can still be dominated by outliers under heavy-tailed noise, potentially degrading the learned vector field. A better discussion of it or a controlled experiment showing robustness under varying noise conditions would strengthen this claim.
> >
> > I will keep my score

---

> > > ### Author Response · Authors · 2026-04-02
> > >
> > > Thank you for clarifying your remaining questions/concerns.
> > >
> > > ## Regarding the identifiability concern:
> > >
> > > 1. The combination of a constrained parameterization (Eq. 1) with the alignment objective (Eq. 6), defines an identifiable mapping $\boldsymbol{\mu}_\psi (\mathbf{x})$ point to point. This induces an identifiable marginal distribution at $\tau=0$.
> > > The compressive flow, via conditional OT bridge construction, learns an identiable transport from $\tau=1$ (original data space) to this already identifiable marginal distribution end-point at $\tau=0$. Therefore, each point in $\tau = 1$ will uniquely (up to sign) transport to $\tau=0$ by the learned compressive flow.
> > >
> > > 2. Moreover, our experiments provide strong empirical support for this claim. In Appendix A.1, despite different random seeds and different penalties on $D$, the latent geometry is preserved up to sign. Further, as explained in our rebuttal to reviewer 2Pev, we further repeated the analysis with 5 random seeds for the mouse video data and again found that DCF recovers the same latent geometry (up to sign flips) and the same four clusters across runs. The mean cosine similarities between independent runs are all extremely close to 1.00 for the 3D latent representation:
> > >
> > > |           | **seed0** | **seed1** | **seed2** | **seed3** | **seed4** |
> > > | --------- | --------: | --------: | --------: | --------: | --------: |
> > > | **seed0** |     1.000 |     0.996 |     0.996 |     0.996 |     0.999 |
> > > | **seed1** |     0.996 |     1.000 |     1.000 |     1.000 |     0.998 |
> > > | **seed2** |     0.996 |     1.000 |     1.000 |     1.000 |     0.997 |
> > > | **seed3** |     0.996 |     1.000 |     1.000 |     1.000 |     0.997 |
> > > | **seed4** |     0.999 |     0.998 |     0.997 |     0.997 |     1.000 |
> > >
> > > We have now also repeated the same analysis on the monkey center-out reach dataset in Section 4.2. Again, DCF shows very high consistency across 5 independent runs, with all off-diagonal cosine similarities between 0.960 and 0.999:
> > >
> > > DCF
> > >
> > > |           | **seed0** | **seed1** | **seed2** | **seed3** | **seed4** |
> > > | --------- | --------: | --------: | --------: | --------: | --------: |
> > > | **seed0** |     1.000 |     0.999 |     0.992 |     0.998 |     0.984 |
> > > | **seed1** |     0.999 |     1.000 |     0.994 |     0.999 |     0.982 |
> > > | **seed2** |     0.992 |     0.994 |     1.000 |     0.996 |     0.960 |
> > > | **seed3** |     0.998 |     0.999 |     0.996 |     1.000 |     0.977 |
> > > | **seed4** |     0.984 |     0.982 |     0.960 |     0.977 |     1.000 |
> > >
> > > By contrast, competing methods (LFADS and CEBRA) are much less stable across seeds:
> > >
> > > LFADS
> > >
> > > |           | **seed0** | **seed1** | **seed2** | **seed3** | **seed4** |
> > > | --------- | --------: | --------: | --------: | --------: | --------: |
> > > | **seed0** |     1.000 |    -0.141 |     0.884 |     0.834 |     0.860 |
> > > | **seed1** |    -0.141 |     1.000 |     0.502 |     0.909 |     0.920 |
> > > | **seed2** |     0.884 |     0.502 |     1.000 |     0.224 |     0.538 |
> > > | **seed3** |     0.834 |     0.909 |     0.224 |     1.000 |     0.958 |
> > > | **seed4** |     0.860 |     0.920 |     0.538 |     0.958 |     1.000 |
> > >
> > > CEBRA
> > >
> > > |           | **seed0** | **seed1** | **seed2** | **seed3** | **seed4** |
> > > | --------- | --------: | --------: | --------: | --------: | --------: |
> > > | **seed0** |     1.000 |     0.718 |     0.373 |     0.428 |     0.769 |
> > > | **seed1** |     0.718 |     1.000 |     0.757 |     0.344 |     0.387 |
> > > | **seed2** |     0.373 |     0.757 |     1.000 |     0.478 |     0.233 |
> > > | **seed3** |     0.428 |     0.344 |     0.478 |     1.000 |     0.478 |
> > > | **seed4** |     0.769 |     0.387 |     0.233 |     0.478 |     1.000 |
> > >
> > > In summary, both principled mathematical argument and empirical findings from three experiments (ball, mouse video and monkey reaching task) demonstrate that DCF reproducibly identifies the same latent space across random seeds and training runs.
> > >
> > > ## Regarding the $L_2$ noise assumption concern:
> > >
> > > The reviewer correctly notes that the standard $L_2$ regression loss can overweight large residuals under heavy-tailed distributions. However, robustness here is influenced not only by the loss, but also by the conditional bridge construction. In particular, recent t-Flow [1] work shows that replacing the conditional Gaussian bridge with a Student-t-type bridge can improve heavy-tail modeling even when the training objective remains L2. We will clarify this point in the revision.
> > >
> > > [1] Pandey, K., Pathak, J., Xu, Y., Mandt, S., Pritchard, M., Vahdat, A., and Mardani, M. Heavy-Tailed Diffusion Models. ICLR 2025

---

### Official Review · Reviewer_2Pev · 2026-03-11

**Soundness:** 3
**Presentation:** 3
**Significance:** 3
**Originality:** 3
**Overall Recommendation:** 4
**Confidence:** 3

**Summary:**

This paper introduces Dynamic Compression Flows (DCFs), a novel dual flow-matching framework that simultaneously learns (1) a compressive flow for dimensionality reduction and (2) a dynamical flow for temporal evolution modeling. The method addresses key challenges in neuroscience data analysis: preserving temporal structure while reducing dimensionality, ensuring latent space identifiability, and handling noise-dominated recordings.

**Compliance With Llm Reviewing Policy:**

Affirmed.

**Final Justification:**

Weak accept.
The paper introduces a technically solid and highly motivated framework that advances the joint modeling of compression and temporal dynamics. The dual-flow formulation is conceptually elegant and achieves strong representational performance.
However, the newly provided runtime comparisons reveal a significant drawback: DCF is substantially more computationally expensive to train than baselines like LFADS. While the framework's capabilities are impressive, this computational overhead remains a practical limitation for high-dimensional or large-scale applications.

**Key Questions For Authors:**

1. Can the authors provide runtime or memory comparisons against baselines such as LFADS, especially since inference requires integrating both flows with a neural ODE solver?
2. The latent-space results are visually appealing, but can the authors provide quantitative metrics for cluster separation, trajectory consistency, or cross-run stability?
3. Is there an optimization trade-off between compression and dynamics preservation? any insight for different data modalities and noise conditions? Since the compressive and dynamical flows are trained jointly, is there evidence that the two objectives do not conflict during optimization?
4. How should practitioners choose the effective latent dimension in practice? as the nested dropout still need input for these hyperparameters. And how this affect to high dimensional data.
5. The method uses linear interpolation bridges. Would nonlinear or geometry-aware bridges improve performance on more strongly nonlinear neural trajectories?

**Limitations:**

yes

**Strengths And Weaknesses:**

Strength
1. The paper is well motivated. The topic about dimensionality reduction and temporal structure preservation have been long debated in neuroscience data. The dual-flow formulation is conceptually clean and extends flow matching to jointly model compression and dynamics.
2. The use of the linear subspace parameterization together with deterministic QR, normalization, and nested dropout is an interesting attempt to improve latent reproducibility and address identifiability issues that are common in other deep latent-variable models.
3. The paper is generally well structured and clearly written, with a good motivation and clear positioning relative to prior flow-based and dynamical approaches.
4. Comprehensive baselines and ablation under different data modalities including synthetic, neural, behavioral, and audio data.

Weakness
1. The identifiability claim should be stated more carefully. The paper gives a plausible structural argument for identifiability up to sign, but the empirical evidence mainly shows reproducibility across runs rather than a stronger demonstration of identifiability in the strict mathematical sense.
2. More quantitative latent-space analysis would strengthen the paper. Much of the evidence for meaningful latent organization is visual, and metrics such as inter-cluster distance, trajectory separation, or stability across runs would make the claim more convincing.
3. The effective latent dimension is still largely controlled by the nested-dropout hyperparameter p, so it is not fully clear how dimension should be selected in practice or how sensitive results are to this choice.
4. The computational cost is not well characterized. Since inference requires integrating both learned flows as neural ODEs, it would be helpful to compare runtime and efficiency with baselines such as LFADS.
5. The interpolation scheme is based on linear bridges in both compression and temporal directions. This is mathematically clean, but it remains unclear how well this approximation captures strongly nonlinear neural dynamics.

---

> ### Author Rebuttal · Authors · 2026-03-30
>
> We'd like to thank the reviewer for their thoughtful comments. Below, we clarify some specific questions/points brought up:
>
> > The identifiability claim should be stated more carefully.
>
> We claim that the encoder head $\mu_\psi$ is identifiable up to a sign. This result follows from the proof in the original nested dropout (ND) paper [1]. In our submission, we gave a brief argument in Section 2.2. In more detail:
> - The encoder is trained with the masked $L_2$ alignment loss (Eq. 6), with a parameterization designed to fix permutation, scaling, and rotation ambiguities. Specifically:
>     - Permutation: nested dropout training of $\mu_\psi$ imposes a prefix-based ordering on the coordinates, so dimensions are unambiguously ordered.
>     - Scaling: Orthonormality of $L$ and normalization of $\mu_\psi$ remove the scale ambiguity between $D$ and $\mu_\psi$.
>     - Rotation: The deterministic construction of orthonormal $L$ through the non-pivoted QR factorization removes arbitrary rotational freedom.
>     - Sign: Sign is not identifiable across runs. Within each run, however, the sign convention is fixed by the non-pivoted QR construction of $L$, with the nonnegative diagonal of $D$.
>
> Our experiments in Appendix A1 empirically verify these claims, and we plan to further clarify this argument in a supplementary note in a revised version of the manuscript.
>
> > Can the authors provide runtime or memory comparisons against baselines such as LFADS ...
>
> The current manuscript lists experiment train times in Appendix B and rollout-time estimates in Appendix C, but we will summarize this more explicitly in the revision and better discuss the computational cost of integrating both flows at inference time. For comparison, the training times of LFADS on the same GPU (in hours) are as follows:
>
> | Dataset | DCF | LFADS  |
> | --- | --- | --- |
> | Ball | 1.97 | 0.045 |
> | Maze | 8.51 | 0.066 |
> | Musall | 5.70 | 0.251 |
>
> > The latent-space results are visually appealing, but can the authors provide quantitative metrics for cluster separation, trajectory consistency, or cross-run stability?
>
> We agree. We repeated our analysis of the video dataset by rerunning DCF, CEBRA, and LFADS for five random seeds. In each case, DCF learned the same latent geometry up to sign flips, and k-means identified the same four clusters as in Figure 4. By contrast, distinct runs of CEBRA revealed similar latent geometry but non-repeatable clustering, while LFADS always resulted in a single, undifferentiated cluster in latent space. In a final version, we plan to repeat this analysis for the full range of comparison models.
>
> > Is there an optimization trade-off between compression and dynamics preservation? ...
>
> We optimize compression and dynamics jointly through the combined objective, setting $\alpha = \beta = \eta = 1$ in most experiments (Appendix A.3 shows that the method is robust to these choices in the rotating-ball case). We did not observe conflict between the two objectives in other experiments, though as compression becomes more severe, we expect tradeoffs with the accuracy of learned dynamics (see next answer).
>
> > How should practitioners choose the effective latent dimension in practice?
>
> Nested dropout controls the latent budget through $p$, where $E(K) = 1/p$. In most of our experiements, we give generous budgets, e.g., p = 1/50, so the model is not tightly constrained a priori. Despite this mild regularization, we found that in practice, the effective dimension $K_{eff}$ (defined as the smallest $K$ explaining 95\% of fitted diagonal mass $\sum_i d_{ii}$) did not strongly depend on the choice of $p$. **Therefore, the practitioner can use $\mathbf{K_{eff}}$ as a post hoc estimate of the effective latent dimension**.
>
> Separately, we can apply a shrinkage loss to $D$, as illustrated in Appendix A.1, to enforce stronger control over the number of dimensions. In some cases, users may intentionally minimize the effective dimensionality, and in these cases, nested dropout provides a soft control on it.
>
> > The method uses linear interpolation bridges. Would nonlinear or geometry-aware bridges improve performance on more strongly nonlinear neural trajectories?
>
> Linear interpolation (conditional OT) bridges usually (not always) lead to the straightest marginal path. The low curvature of the path makes training easier, makes data generation more efficient, and enhances the sample quality.
>
> In addition:
> - We use small timesteps for integration, for which the assumption of conditional linear interpolation is comparatively mild.
> - While conditional OT interpolation assumes _conditional_ linearity, it does not enforce _marginal_ linearity. The marginal distributions at either end of the flow are undistorted by this assumption.
>
> [1] Rippel, O., Gelbart, M., and Adams, R. P. (2014). Learning Ordered Representations with Nested Dropout. Proceedings of the 31st International Conference on Machine Learning (ICML), PMLR 32(2):1746-1754.

---

> > ### Author Rebuttal · Reviewer_2Pev · 2026-04-02
> >
> > Thank you for the detailed clarifications regarding identifiability, runtime comparisons, the planned addition of quantitative latent-space analysis in the revision, and the methodological choices overall. These responses addressed several of my concerns. I think some of these points should be stated more explicitly in the final version and I will keep my current score.

---

### Official Review · Reviewer_Rvy8 · 2026-03-13

**Soundness:** 3
**Presentation:** 3
**Significance:** 2
**Originality:** 2
**Overall Recommendation:** 3
**Confidence:** 4

**Summary:**

In terms of the neural and behavioral time-series data by learning low-dimensional representations that preserve temporal dynamics, the authors developed a dual flow-matching approach using two flow fields. One governs time evolution in the latent space, and the other defines a mapping between the original data and the latent space. The dimension-reducing flow is trained to minimize distortions of temporal dynamics, ensuring that temporal relationships in the original data are preserved in the latent representation. The method is tested on both neural and behavioral data, producing more interpretable dynamics and higher-quality reconstructions compared to existing models.

**Compliance With Llm Reviewing Policy:**

Affirmed.

**Final Justification:**

The authors addressed some of my concerns, so I will follow my initial review, staying neutral for this paper.

**Key Questions For Authors:**

1, Your current formulation is based on flow-based generative models. How could your method be extended to diffusion-based models? Would this require significant changes to your model architecture and implementation? Given the strong competitors in the generative modeling space—such as rectified flow, diffusion probabilistic models, and score-based drifting models—can your approach be adapted to or combined with these alternative frameworks? If so, what would be your methodological edge over them?
2, How well does your approach generalize across different neural network architectures, such as Transformers and U-Nets? Are there significant differences in performance or training dynamics depending on the underlying architecture? If so, what are the key factors driving those differences?
3, It seems that your method, Dynamic Compression Flow, is a combination of flow matching techniques with other neural network components. Are there novel contributions to the flow matching methodology itself, beyond the integration with dimensionality reduction and temporal dynamics preservation?

**Limitations:**

yes

**Strengths And Weaknesses:**

This paper presents a well-written study that addresses an important problem in neuroscience: learning low-dimensional latent representations that preserve temporal dynamics. The authors' motivation is compelling, and their dual flow-matching framework offers a solution to the distortions introduced by static dimensionality reduction methods.

However, the method mostly combines existing components—a temporal flow and a dimension-reducing flow—without more insightful innovations. From another point of view, the paper would benefit from a more thorough engagement with the rapidly evolving landscape of generative models: diffusion-based, drifting, etc. Given the emergence of powerful alternatives such as rectified flow and score-based drifting models, it remains unclear how the proposed method compares against these stronger baselines. While I think it would be better to submit this paper to more area-specific conferences or journals, I would like to rate this paper as neutral since the studied scenario and method are indeed interesting.

---

> ### Author Rebuttal · Authors · 2026-03-30
>
> We thank the reviewer for their thoughtful comments. In response to the weaknesses noted by the reviewer:
>
> > Your current formulation is based on flow-based generative models. How could your method be extended to diffusion-based models? ... If so, what would be your methodological edge over them?
>
> Gaussian Flow Matching and Diffusion Models can be viewed within a common framework, wherein the velocity field is linked to the score function by reparameterization. In particular, when the source distribution is a standard Gaussian, flow matching and many diffusion models admit a unified view. This ICLR blog [1] provides a very clear discussion of this connection.
>
> Flow matching (FM) often shows advantages in efficiency and sample quality when compared to many diffusion-based generative models. Moreover, there are at least two specific reasons why we chose FM for this work:
> 1. Flow matching allows us to capture transitions between arbitrary source and target distributions, while most diffusion models are restricted to a Gaussian source. This is particularly important because our neural dynamics flow must link non-Gaussian marginal activity distributions across time in data.
>
> 2. Flow matching allows us to learn flow fields directly from data, rather than implicitly through the score function, which presents numerical difficulties. In particular, the dynamics flow field reveals how neural activity evolves through time and is linked, in the "computation through dynamics" paradigm, to neural computation. Thus, our model directly learns a velocity field that is of interest to neuroscientists in its own right.
>
> All of which is to say that flow matching not only offers us good generative performance; it also allows for the _inference_ of flow fields that can be used for the study of brain dynamics.
>
> > How well does your approach generalize across different neural network architectures, such as Transformers and U-Nets?
>
> Theoretically, one can use any architecture to model the velocity/ score component in the flow matching/diffusion-based model framework. Pragmatically, practitioners often choose different models/architectures (e.g., Transformers, U-Nets and tree-based models for tabular data [2]) based on the specific problem they are interested in. In our experiments, we used MLPs for neural spike data and U-Nets for image data.
>
> >  It seems that your method, Dynamic Compression Flow, is a combination of flow matching techniques with other neural network components. Are there novel contributions to the flow matching methodology itself, beyond the integration with dimensionality reduction and temporal dynamics preservation?
>
> Neural networks are the standard way to parameterize the velocity field in flow matching. Our contribution is not a new approach to flow matching, but we are the first, to our knowledge, to extend flow matching to the simultaneous learning of multiple flow fields, corresponding to multiple independent types of data transformation. We are also the first to combine flow matching with dimension reduction _in an **identifiable** way_, allowing for _reproducible_ inference of low-dimensional neural dynamics. Again, to our knowledge, these features of our flow matching setup are both methodologically novel.
>
> [1] Gao, R., Hoogeboom, E., Heek, J., De Bortoli, V., Murphy, K. P., and Salimans, T. Diffusion Models and Gaussian Flow Matching: Two Sides of the Same Coin. ICLR 2025 Blogposts Track (https://d2jud02ci9yv69.cloudfront.net/2025-04-28-diffusion-flow-173/blog/diffusion-flow/)
>
> [2] Jolicoeur-Martineau, A., Fatras, K., and Kachman, T. Generating and Imputing Tabular Data via Diffusion and Flow-based Gradient-Boosted Trees. Proceedings of AISTATS 2024, PMLR 238:1288-1296.

---

> > ### Author Rebuttal · Reviewer_Rvy8 · 2026-04-02
> >
> > I thank the authors for their response, which addresses some of my concerns. I will follow my initial review, staying neutral for this paper.

---

### Official Review · Reviewer_HAqv · 2026-03-13

**Soundness:** 2
**Presentation:** 3
**Significance:** 2
**Originality:** 3
**Overall Recommendation:** 5
**Confidence:** 4

**Summary:**

In this work, the authors proposed a framework named Dynamic Compression Flow, which is a continous density transfomation method that temporally redistributes the inferred latent representations. Specifically, the authors employ a two-stage flow fields approach, the first flow is used to model the time evolution, while the seocnd is used to learn the mapping between the high-dimensional data and the low-dimensional latent space. The experimental results are conducted on both neural and behavioral data, which produces interpretable neural latent dynamics, as well as high-reconstruction quality.

**Compliance With Llm Reviewing Policy:**

Affirmed.

**Final Justification:**

I thank the authors for the response. The detailed comparison between the proposed method based on FM and diffusion models make sense to me. I will raise my score to 5.

**Key Questions For Authors:**

I have no more questions, for my concerns, please refer to my 'weaknesses' section.

**Limitations:**

yes.

**Strengths And Weaknesses:**

Strengths:

* I believe that the use of powerful flow matching is a great step for more advanced and expressive modeling of the latent density distribution.
* In Section 2.3, the design of the compression flow and the compression velocity field is helpful to promote the model learning how the latent representations are varying on the path axis.
* The figures in the paper are well-illustrated and easy-to-follow.



Weaknesses:

* While diffusion models can be viewed as a special case of flow-matching models, works on diffusion models [1], [2] applied to neuroscience and neural data are not discussed and compared in this work.
* For Table 1 in the experiment section, does this is a fair comprison between all the methods? Do all the methods expose to the same amount of neural data time window and w/o any labeled behavior data? Why some models performed almost ~0 in the explained variance values?





[1] Extraction and Recovery of Spatio-Temporal Structure in Latent Dynamics Alignment with Diffusion Models. NeurIPS 2023, Wang, et, al.

[2] Latent Diffusion for Neural Spiking Data. NeurIPS 2024, Kapoor, et, al.

---

> ### Author Rebuttal · Authors · 2026-03-30
>
> Thank you for your thoughtful comments. In reply to the specific points raised by the reviewer:
>
> > While diffusion models can be viewed as a special case of flow-matching models, works on diffusion models [1], [2] applied to neuroscience and neural data are not discussed and/or compared in this work.
>
> Thank you for pointing out these related lines of work. We regret the omission in our submission. We will both cite and discuss the similarities and differences between these methods and ours in the final manuscript.
> Briefly, when compared to these two competing approaches, our dynamic compressive flows method focuses on learning identifiable low-dimension representations that preserve the temporal structure. That is, we focus on inference, even though our approach also preserves generative capabilities of flow matching (FM) models. Moreover, the velocity fields learned via FM are themselves biologically informative. Please refer to our reply to Reviewer Rvy8 for details on the relationship between FM and diffusion-models as well as our motivation to choose FM instead of diffusion-models in this work.
>
> > For Table 1 in the experiment section, does this is a fair comparison between all the methods? Do all the methods expose to the same amount of neural data time window and w/o any labeled behavior data?
>
> Table 1 reports held-out linear decoding of cursor velocity from latent representations ($d=3$) for all models on the same center-out reaching dataset. In this experiment, the neural data consist of 592 training trials and 197 held-out test trials, with smoothed spike counts from 137 neurons over 100 time steps. The goal here is to test whether different latent representations preserve behaviorally relevant structure under the same downstream decoding task. For this analysis all methods were exposed to the same amount of data and time window, as explained above, without any additional trial information. See methods for additional details.
>
> > Why some models performed almost ~0 in the explained variance values?
>
> Both DMD and T-PHATE performed poorly on this task, and we believe these are for different reasons. For T-PHATE, it is apparent in Figure S5 that the learned latent space fails to capture the latent structure of the task, so poor decoding performance there is unsurprising. For DMD, the fact that we did not use a delay embedding to expand the task dimension prior to modeling likely hurt the ability of the model, which is linear, to accurately capture dynamics. That other nonlinear methods, including DCF, performed well in this task suggest that nonlinearity is essential to reasonable decoding performance.

---

> > ### Author Rebuttal · Reviewer_HAqv · 2026-04-03
> >
> > I thank the authors for the response. The detailed comparison between the proposed method based on FM and diffusion models make sense to me. I will raise my score to 5.

---

### Decision · Program_Chairs · 2026-04-30

**Decision:**

Accept (regular)

**Comment:**

This paper’s starting point is the puzzling observation in neuroscience that, on the one hand, large populations of neurons are responsive in any one task, on the other many studies report neural dynamics to be confined to low-dimensional manifolds. The present work aims to achieve both, a faithful representation of the latent dynamics yet on a low-dimensional manifold that respects the system’s temporal evolution. With that, it addresses a particularly important problem in neuroscience.

The paper received mixed reviews, with two accepts and two rejects. The basic flow matching idea appears quite nice and potentially powerful to me, although flow matching per se and learning a low-d embedding jointly with dynamics are in themselves not very novel contributions (cf. Ref. mcAW). Another strong point of the paper is the consideration of a set of quite diverse real-world datasets (neural, video of behavior, and audio of bird songs), highlighting the wide applicability. Reproducibility and identifiability issues, which plague many models in neuroscience, are also partially addressed. While not all referees were fully convinced regarding the point about identifiability, reproducibility was further supported by additional experiments during the rebuttal. Another weakness that was brought up is the high computational costs, actually reflected in new experiments in response to Ref. 2Pev.

The methodology may be seen more as providing a low-d representation of the data that respects their temporal dynamics, rather than yielding a *generative* model of the dynamics in low-d (at least that was not explicitly shown). This curtails its usefulness a bit. The baselines used (LFADS, CEBRA, VAR …) also either seem quite basic or a bit outdated to me (not reflecting SOTA anymore, at least not in dynamical systems). But the basic idea may have some value particularly in neuroscience.